# A genetic particle filter scheme for univariate snow cover assimilation into Noah-MP model across snow climates

Yuanhong You[1,3], Chunlin Huang[2], Zuo Wang[1], Jinliang Hou[2], Ying Zhang[2], Peipei Xu[1]

[1]Anhui Normal University, School of Geography and Tourism, Key Laboratory of Earth Surface Processes and Regional Response in the Yangtze-Huaihe River Basin of Anhui Province, Wuhu, 241002, China

[2]Northwest Institute of Eco-Environment and Resources, Chinese Academy of Sciences, Lanzhou, 730000, China

[3]Engineering Technology Research Center of Resource Environment and GIS, Wuhu, 241002,China

Corresponding author: Chunlin Huang, Key Laboratory of Remote Sensing of Gansu Province, Northwest Institute of Eco-Environment and Resources, Chinese Academy of Sciences, Lanzhou, Gansu, 730000, China. (huangcl@lzb.ac.cn)

# Abstract

Accurate snowpack simulations are critical for regional hydrological predictions, snow avalanche prevention, water resource management, and agricultural production, particularly during the snow ablation period. Data assimilation methodologies are increasingly being applied for operational purposes to reduce the uncertainty in snowpack simulations and enhance their predictive capabilities. This study aims to investigate the feasibility of using Genetic Particle Filter (GPF) as a snow data assimilation scheme designed to assimilate ground-based snow depth (SD) measurements across different snow climates. We employed the default parameterization scheme combination within the Noah-MP model as the model operator in the snow data assimilation system to evolve snow variables and evaluated the assimilation performance of GPF using observational data from sites with different snow climates. We also explored the impact of measurement frequency and particle number on the filter updating of the snowpack state at different sites and the results of generic resampling methods compared to the genetic algorithm used in the resampling process. Our results demonstrate that GPF can be used as a snow data assimilation scheme to assimilate ground-based measurements and obtain satisfactory assimilation performance across different snow climates. We found that particle number is not crucial for the filter's performance, and 100 particles are sufficient to represent the high dimensionality of the point-scale system. The frequency of measurements can significantly affect the filter updating performance, and dense ground-based snow observational data always dominate the accuracy of assimilation results. Compared to generic resampling methods, the genetic algorithm used to resample particles can significantly enhance the diversity of particles and prevent particle degeneration and impoverishment. Finally, we concluded that the GPF is a suitable candidate approach for snow data assimilation and is appropriate for different snow climates.

# 1. Introduction

Understanding snowpack dynamics is crucial for water resource management, agricultural production, avalanche prevention and flood preparedness in snow dominated regions (Piazzi et al., 2019; Pulliainen et al., 2020). As a special land surface type, seasonal snow cover is highly sensitive to climate change and has a significant impact on energy and hydrological processes (Barnett et al., 2005; Takala et al., 2011; Kwon et al., 2017; Che et al., 2014). On one hand, the high albedo of snow-covered surfaces can significantly reduce shortwave radiation absorption, leading to adjustments in the energy exchange between the land surface and atmosphere (You et al., 2020a; You et al., 2020b). On the other hand, the low thermal conductivity of snow cover can insulate the underlying soil, which results in reduced temperature variability and a more stable environment (Zhang et al., 2005; Piazzi et al., 2019). In addition, snowmelt is a vital source of water that plays a critical role in soil moisture,

runoff, and groundwater recharge (Dettinger, 2014; Griessinger et al., 2016; Oaida et al., 2019). Therefore, comprehending snow dynamics is essential for predicting snowmelt runoff, atmospheric circulation, hydrological predictions, and climate change.

Currently, there is a growing effort to investigate the potential of data assimilation (DA) schemes to improve snow simulations and obtain the optimal posterior estimate of the snowpack state (Bergeron et al., 2016; Piazzi et al., 2018; Smyth et al., 2020; Abbasnezhadi et al., 2021). Various DA methodologies with different degrees of complexity have been developed, resulting in diverse performance levels. Sequential DA techniques, including basic direct insertion, optimal interpolation schemes, ensemble-based Kalman filter, and particle filter, have been widely employed in real-time applications. The greatest strength of sequential DA techniques is that the model state can be sequentially updated when observational data become available (Piazzi et al., 2018). However, the direct insertion method, which replaces model predictions with observations when available, is based on the assumption that the observation is perfect and the model prior is wrong (Malik et al., 2012). This method can potentially result in model shocks due to physical inconsistencies among state variables (Magnusson et al., 2017). Although the optimal interpolation method is more advanced and takes into account observational uncertainty, it still has great limitations and is rarely used in real-time operational systems (Dee et al., 2011; Balsamo et al., 2015).

At a higher level are the Kalman filter and ensemble-based Kalman filter, which are most commonly used in various real-time applications. The Ensemble Kalman Filter (EnKF), which was first introduced by Evensen in 2003, uses a Monte Carlo approach to approximate error estimates based on an ensemble of model predictions. This approach does not require model linearization, making it particularly advantageous. Precisely due to this advantage, the EnKF has been widely used in snowpack prediction. For example, EnKF has been used to assimilate MODIS snow cover extent and AMSR-E SWE into a hydrological model to improve modeled SWE (Andreadis et al., 2006), as well as to assimilate MODIS fractional snow cover into a land surface model (Su et al., 2008). Moreover, the EnKF method has been used to enhance snow water equivalent estimation by assimilating ground-based snowfall and snowmelt rates, assimilation of both D-InSAR (Differential Interferometric Synthetic Aperture Radar) and manually measured snow depth data simultaneously (Yang and Li, 2021). Even though there are numerous studies have generally stated that the EnKF has an excellent assimilation performance enabling it to consistently improve snow simulations, some constraining limitations hinder the filter performance (Chen, 2003). One of the main limitations is that the EnKF assumes that the model states follow a Gaussian distribution and only considers the first and second order moments, thereby losing relevant information contained in higher-order moments (Moradkhani et al., 2005). Unfortunately, the dynamical system usually has strong nonlinearity and the involved probability distribution of system state variables is not supposed to follow a Gaussian distribution (Weerts and El Serafy, 2006). Additionally, the filter performance of

the EnKF is significantly influenced by the linear updating procedure, and the state-averaging operations can be particularly challenging for highly detailed complex snowpack models.

In order to overcome these limitations, the particle filter (PF) which also based on Monte Carlo method has been developed for non-Gaussian, nonlinear dynamic models (Gordon et al., 1993). The greatest strength of PF technique is to be free from the constraints of model linearity and error following a Gaussian distribution. This enables the successful application of the PF technique to nonlinear dynamical systems with non-Gaussian errors. Additionally, the PF technique gives weights to individual particles but leave model states untouched, which makes PF more computationally efficient than the ensemble Kalman filter and smoother techniques (Margulis et al., 2015). Thanks to these advantages, an increasing interest focuses on applying PF technique in snow data assimilation. For example, remotely sensed microwave radiance data were assimilated into a snow model to update model states using the PF technique, and the results demonstrated that the SWE simulations have great improvement (Dechant and Moradkhani, 2011; Deschamps-Berger et al., 2022). A newly PF approach proposed by Margulis et al. (2015) was used to improve SWE estimation through assimilating remotely sensed fractional snow-covered area. At basin scale, PF technique was implemented with the objective of obtaining high resolution retrospective SWE estimates (Cortes et al., 2016). The PF technique was also used to assimilate daily snow depth observations within a multi-layer energy-balance snow model to improve SWE and snowpack runoff simulations (Magnusson et al., 2017). The studies indicated above demonstrated that the assimilated snow-related in-situ measurements or the remotely sensed observation data through the PF technique can successfully update predicted snowpack dynamics, and the PF scheme is a well-performing data assimilation technique enabling to consistently improve model simulations. Nevertheless, particle degeneracy is still a potential limitation of the PF technique. It occurs when most particles have negligible weight, and only a few particles carry significant weights, which hinders a realistic sampling of the underlying probability distribution of the state (Parrish et al., 2012; Abbaszadeh et al., 2017; Abbaszadeh et al., 2018). The particle resampling has been considered to be an efficient approach that can effectively mitigate the problem of particle degeneracy. However, it may result in a sample containing many repeated points and a lack of diversity among the particles, which is referred to as sample impoverishment (Rings et al., 2012; Zhu et al., 2018). And the sample impoverishment was a tricky problem for generic resampling methods. Using intelligent search and optimization methods to mitigate the degeneracy problem may be a good choice because it can effectively avoid sample impoverishment (Park et al., 2009; Ahmadi et al., 2012; Abbaszadeh et al., 2018). The Genetic Algorithm (GA) as an intelligent search and optimization method has been known as an effective approach to mitigate the degeneracy problem and received more attention (Kwok et al., 2005; Park et al., 2009; Mechri et al., 2014). The GA applied in the particle filter, which is referred to as the genetic particle filter (GPF), has been successfully implemented to estimate parameters or states in

nonlinear models (Van Leeuwen, 2010; Snyder, 2011). The GPF was also used as data assimilation scheme applied to land surface model which simulates prior subpixel temperature and the results showed the GPF outperformed prior model estimations (Mechri et al., 2014). Despite a series of studies having proven that the GPF is an effective data assimilation approach, however, few studies have investigated the performance of GPF as a snow data assimilation scheme, especially in different snow climates. In view of the promising performances of GPF as a snow data assimilation scheme, this paper aims to investigate the potential of GPF in performing snow data assimilation, and the main goal of this research is to address the following issues: (1) Can the GPF be employed as a snow data assimilation scheme? (2) How is the assimilation performance of GPF in snow data assimilation across different snow climates? (3) The sensitivity of DA simulations to the frequency of the assimilated measurements and the particle number.

This paper is organized as follows. Section 2 introduces the study sites, the meteorological dataset, the snow module within the Noah-MP model, the calculation flow of the GPF scheme, and design of the numerical experimental. Section 3 explains the simulation results of SD using the open-loop ensemble, explores the sensitivity of the measurement frequency and ensemble size. Finally, section 4 summarizes the findings of this study.

## 2. Materials and methods

### 2.1 Study sites and data

With consideration of the filtering performance, which may vary in snow climates, eight seasonally snow-covered study sites with different snow climates were selected to implement numerical experimental in this study (Sturm et al., 1995; Trujillo and Molotch, 2014). These sites are distributed at different latitudes in the Northern Hemisphere, and the sites included the Arctic Sodankylä site (SDA, 179 m), located beside the Kitinen River in Finland and the upper 2 meters are frozen (Rautiainen et al., 2014); the Snoqualmie site (SNQ, 921 m) with a rain-snow transitional climate in the Washington Cascades of the USA, the SD measured by snow stakes was employed (Wayand et al., 2015); the maritime Col de Porte (CDP, 1330 m) site in the Chartreuse Range in the Rhone-Alpes region of France; the Mediterranean climate Refugio Poqueira site (ROPA, 2510 m) in Sierra Nevada Mountains of Spain and has a high evaporation rate (Herrero et al., 2009); the Weissfluhjoch site (WFJ, 2540 m) in Davos of Switzerland, and automatic SD observations used in this study (Wever et al., 2015); the continental Swamp Angel Study Plot (SASP, 3370 m) site in the San Juan Mountains of Colorado, USA; and two sites from typical snow-covered regions in China, the Altay meteorological observation site (ATY, 735.3 m) in Northern Xinjiang, China, where there is less wind in the winter season; the other one is the Mohe meteorological observation site (MOHE, 438.5 m) in a county of Northeast China, which has a cold temperate continental climate and is the

northernmost part of China. Serially complete meteorological measurements are available and can be used as forcing data in these sites, certainly, the downward longwave and shortwave radiation values of MOHE were extracted from the China Meteorological Forcing Dataset (CMFD) (Chen et al, 2011), since there are no radiation measurements in this site.

It is noteworthy that the spatial variance of the performance of the model is negligible since these sites themselves are flat and the surrounding vegetation types are uniform. We have used this data set to examine the sensitivity of simulated SD to physics options, and the results shown that the dataset has a reliable quality. In addition, the location, the detailed information of snow climates, and details about the dataset processing for the eight sites can be also referenced in You et al. (2020a).

## 2.2 Snow module within Noah-MP model

The snow partial module within Noah-MP model can be divided into up to three layers, depending on the depth of the snow (Yang et al., 2011). The SD $h_{snow}$ is calculated by

$$h_{snow}^t = h_{snow}^{t-1} + \frac{P_{s,g}}{\rho_{sf}} dt \; . \tag{1}$$

where $P_{s,g}$ is the snowfall rate at the ground surface, $dt$ is the timestep, and $\rho_{sf}$ is the bulk density of the snowfall. When $h_{snow} < 0.025\,\mathrm{m}$, the snowpack is combined with the top soil layer, and no dependent snow layer exists. When $0.025 \leq h_{snow} \leq 0.05\,\mathrm{m}$, a snow layer is created with a thickness equal to SD. When $0.05 < h_{snow} \leq 0.1\,\mathrm{m}$, the snowpack will be divided into two layers, each with a thickness of $\Delta z_{-1} = \Delta z_0 = h_{snow}/2$. When $0.1 < h_{snow} \leq 0.25\,\mathrm{m}$, the thickness of the first layer is $\Delta z_{-1} = 0.05\,\mathrm{m}$, and the thickness of the second layer is $\Delta z_0 = (h_{snow} - \Delta z_{-1})\,\mathrm{m}$. When $0.25 < h_{snow} \leq 0.45\,\mathrm{m}$, a third layer is created, and the three thickness are: $\Delta z_{-2} = 0.05\,\mathrm{m}$ and $\Delta z_{-1} = \Delta z_0 = (h_{snow} - \Delta z_{-2})/2\,\mathrm{m}$. When $h_{snow} > 0.45\,\mathrm{m}$, the layer thickness of the three snow layers are $\Delta z_{-2} = 0.05\,\mathrm{m}$, $\Delta z_{-1} = 0.2\,\mathrm{m}$, $\Delta z_0 = (h_{snow} - \Delta z_{-2} - \Delta z_{-1})\,\mathrm{m}$. Certainly, the snow cover is highly influenced by air and ground temperature, and the snow layer combines with the neighboring layer due to sublimation or melting and is redivided depending on the total SD. The snow module of the Noah-MP model provides an estimate of snow-related variables using energy and mass balance. This computing process requires a series of meteorological forcing data, such as near-surface air temperature, precipitation, and downward solar radiation. The snow accumulation or ablation parameterization of the Noah-MP model is based on the mass and energy balance of the snowpack, and the snow water equivalent can

be calculated using the following equation:

$$\frac{dW_s}{dt} = P_{s,g} - M_s - E.$$ (2)

where $W_s$ is the snow water equivalent (mm), $P_{s,g}$ is the solid precipitation (mm s$^{-1}$), $M_s$ is the snowmelt rate (mm s$^{-1}$), $E$ is the snow sublimation rate (mm s$^{-1}$).

A snow interception model was implemented into the Noah-MP model to describe the process of snowfall intercepted by the vegetation canopy (Niu and Yang, 2004). Within this model, the snowfall rate at the ground surface $P_{s,g}$ is then calculated by

$$P_{s,g} = P_{s,drip} + P_{s,throu}.$$ (3)

where $P_{s,drip}$ (mm s$^{-1}$) is the drip rate of snow and $P_{s,throu}$ (mm s$^{-1}$) is the through-fall rate of snow. In the Noah-MP model, the ground surface albedo is parameterized as an area-weighted average of the albedos of snow and bare soil, and the snow cover fraction of the canopy is used to calculate the ground surface albedo, as shown in Equation (4),

$$\alpha_g = \left(1 - f_{snow,g}\right)\alpha_{soil} + f_{snow,g}\alpha_{snow}.$$ (4)

where $\alpha_{soil}$ and $\alpha_{snow}$ are the albedo of bare soil and snow, respectively. $f_{snow,g}$ is the snow cover fraction on the ground and is parameterized as a function of snow depth, ground roughness length, and snow density (Niu and Yang, 2006).

## 2.3 Genetic particle filter data assimilation scheme

The Bayesian recursive estimation problem is solved by the Monte Carlo approach within PF technique, making this scheme appropriate for nonlinear system with a non-Gaussian probability distribution (Magnusson et al., 2017). The basic concept of PF technique is to use a large number of randomly generated realizations (i.e., particles) of the system state to represent the posterior distribution. Meanwhile, the particles are propagated forward in time as the model evolves. The weights associated with the particles are updated based on the likelihood of each particle's simulated proximity to the real observation. The weight of the particles can be updated as follows:

$$w_t^i = w_{t-1}^i p\left(z_t \middle| x_t^i\right).$$ (5)

where $w_{t-1}^i$ is the weight of $i$ th particle at time $t-1$ and the weight is updated by the likelihood function $p\left(z_t \middle| x_t^i\right)$, which measures the likelihood of a given model state with respect to the observation $z_t$. The observation errors are generally assumed to follow a Gaussian distribution, and

the chosen likelihood function represents this assumption. In this study, we employed a normal

probability distribution to serve as likelihood function:

$$p\left(z_t \middle| x_t^i\right) = N\left(z_t - x_t^i, \sigma\right). \tag{6}$$

where $N$ represents the normal probability distribution of the residuals between observed, $z_t$, and

simulated, $x_t$. Finally, the weights of the updated model state would be normalized, and the

assimilated value of model state is the weighted average of all particles at time $t$. Although the

particle filter has been widely applied in various nonlinear systems, the particle degeneracy and

impoverishment in particle filter are still the fatal limitations need to be urgently addressed. To

address the degeneration problem in PF technique, traditional resampling methods like multinominal

resampling, systematic resampling were employed to resample the particles if the effective sample

size,

$$N_{eff} = 1 \middle/ \sum_{i=1}^{N} \left(w_t^j\right)^2. \tag{7}$$

fell below a specified number. Where $N$ is the ensemble size and $w_t^j$ is the normalized weights

defined in Equation (5). To be honest, traditional resampling methods can effectively mitigate the

problem of particle degeneracy by resampling high-quality particles. However, after multiple

iterations, these methods often lead to a serious lack of diversity among particles, which is known as

the particle impoverishment problem. To mitigate both of these issues simultaneously, we employed

the genetic algorithm (GA) to resample the particles, resulting in the genetic particle filter algorithm

(GPF). The GA is inspired by Darwin's theory of evolution and emphasizes the principle of survival

of the fittest. In fact, in the resampling phase, the fitness of particles should be reselected according

to the theory of particle filtering. Selection, crossover, and mutation are major steps used to simulate

population evolution. As shown in Figure 1, these three operators are utilized to produce better

offspring and improve the overall population fitness, with the aim of preventing particle degeneracy

and impoverishment. These operators will be used to improve particle fitness when it falls below a

threshold value. The three operators are described below.

**Selection mechanism**: At the time of assimilation, the selection operator will preferentially select the

particles that are close to the observed SD. This process is usually achieved by sorting the fitness

value of all particles and selecting a certain proportion of particles. Here, we calculated the survival

rate of all individuals and sorted them in ascending order. The top fifth percentile of particles were

considered high-quality particles and were selected as parents in genetic algorithm. This ensures that

fit individuals can be delivered to the next generation group. The survival rate of particles can be

calculated using the following equation:

$$P(x_{t,i}) = \exp\left[-\frac{1}{R_k}\left(x_{i,k|k-1} - z_k\right)^2\right].\tag{8}$$

where $R_k$ is the observation error at time $k$, 0.01 m was set in this study; $z_k$ represents the observed SD.

**Crossover mechanism**: The purpose of crossover operator is to exchange some genes for two or more chromosomes in a specified way, creating new individuals. GA mainly generates new individuals through this process, which determines the capability of global search. In this study, the arithmetic crossover method was used as the crossover operator to generate new individuals. Two particles were randomly selected from the resampled particle group and combined linearly to form a new particle. Assuming the two selected particles are $\{x_m, x_n\}$, the following equations were used to form the new particles:

$$x_m^{'} = \alpha x_m + (1-\beta)x_n.\tag{9}$$

$$x_n^{'} = \beta x_n + (1-\alpha)x_m.\tag{10}$$

where $\alpha$, $\beta$ are the empirical crossover coefficients, and $\alpha = 0.45$, $\beta = 0.55$ in this study. In order to ensure diversity among particles, newly formed particles will be discarded when the $x_m^{'} = x_n^{'}$ occurred, and parent individuals will be re-selected from the particle group.

**Mutation mechanism**: The mutation in GA refers to replacing the gene values at some loci with other alleles to form a new individual. The mutation mechanism can be considered as a supplement to the crossover mechanism, which can increase the diversity of the population. Assuming that the randomly selected particle from the crossed particle set is $x_k$, the mutation operation is performed on the particle using the following equation:

$$x_k^{'} = x_k + \eta * Uniform.\tag{11}$$

where $Uniform$ refers a random number from a uniform distribution, $\eta$ is an empirical coefficient, and 0.01 was set in this study.

It is noteworthy that a large number of particles may lead to filter collapse. In this study, we set the number of particles equal to 100 based on previous references (Mechri et al., 2014; Magnusson et al., 2017; Piazzi et al., 2018). Moreover, to prevent the particle ensemble from being unable to represent the prior model state due to structural deficiencies, a Gaussian-type model error, $N(\mu, \sigma)$, was added to the ensemble members. The $\mu$ was obtained from the mean value of residual between

simulation and observation, and the variance $\sigma$ was set to 0.01.

## 2.4 DA experimental design

### 2.4.1 Perturbation of meteorological input data

The accuracy of models' output largely depends on the input meteorological forcing dataset for
land surface models, and meteorological forcing are one of the major sources of uncertainty affecting
simulation results (Raleigh et al., 2015). The precipitation and air temperature are the most important
input elements for snow simulations since their roles in determining the quantity of rainfall and
snowfall.

To produce the forcing data ensemble, the air temperature and precipitation were perturbed
following the method of Lei et al. (2014). In this study, the precipitation was assumed to have an error
with a log-normal distribution, and it is expressed as follows:

$$P_t^i = \exp\left(\mu_{\ln P} + \varphi_{P,i} \cdot \sigma_{\ln P} / 2\right). \tag{12}$$

$$\sigma_{\ln P} = \sqrt{\ln\left(\frac{\left(\alpha_p \cdot P_t\right)^2}{P_t^2} + 1\right)}. \tag{13}$$

$$\mu_{\ln P} = \ln\left(\frac{P_t^2}{\sqrt{P_t^2 + \left(\alpha_p \cdot P_t\right)^2}}\right). \tag{14}$$

where $P_t$ and $P_t^i$ are the observed and perturbed precipitation at time $t$, respectively. The log

transformation of $P_t^i$ is a Gaussian distribution with a mean ($\mu_{\ln P}$) and a standard deviation ($\sigma_{\ln P}$);

$\alpha_P$ is the variance scaling factor of the precipitation, which was set to 0.5 in this study; and $\varphi_{P,i}$ is

a normally distributed random number. Meanwhile, the ensemble of the air temperature was obtained
as follows:

$$T_t^i = T_t - \gamma\left(1 - 2w^i\right), w^i \sim U\left(0,1\right). \tag{15}$$

Where $T_t$ and $T_t^i$ are the observed and perturbed air temperatures at time $t$, respectively; $\gamma$

is the variance scaling factor of the temperature with a value of 2.0; and $w^i$ is the random noise with

a uniform distribution between 0 and 1. A forcing ensemble containing 100 particles was obtained
through above perturbation method in this study.

### 2.4.2 Evaluation metrics

In order to properly quantify the filter performance, each experiment is evaluated by statistical

analysis based on the daily mean values of simulations and observations. In this study, we used the Kling-Gupta efficiency (KGE) coefficient (Gupta et al., 2009) to evaluate the filter performance, which allows the analysis of how the assimilation of snow observations succeeds in properly updating the model simulations, on average:

$$KGE = 1 - \sqrt{(r-1)^2 + (a-1)^2 + (b-1)^2} \ . \tag{16}$$

where $r$ is the linear correlation coefficient between the simulated and observed SD; $a$ is the ratio of the standard deviation of simulated SD to the standard deviation of the observed ones; and $b$ is the ratio of the mean of simulated SD to the mean of observed ones, here, the simulated SD is the mean SD ensemble simulations. Theoretically, when $r = 1$, $a = 1$ and $b = 1$ in Equation (16), the KGE will obtain the optimal value which equals to 1, and this illustrates that the simulated SD highly consistently with the observed ones.

The time series of SD obtained from assimilation scenarios was compared to observations for evaluating the performance of the assimilation, and the root-mean-square error (RMSE) was employed:

$$RMSE = \sqrt{\frac{1}{N} \sum_{i=1}^{N} (obs(i) - sim(i))^2} \ . \tag{17}$$

where $N$ is the total number of observations, $sim(i)$ is the simulated value at time $i$, and $obs(i)$ is the observed value at time $i$.

Another statistical index is the continuous ranked probability skill score (CRPSS), which is evaluated to assess changes to the overall accuracy of the ensemble simulations of each experiment (CRPS) by considering the open-loop ensemble control run as the reference one ($CRPS_{ref}$), and the calculation scheme is shown in the following formula:

$$CRPSS = 1 - \frac{CRPS}{CRPS_{ref}} \ . \tag{18}$$

where CRPS is the continuous ranked probability score which can measure the difference between continuous probability distribution and deterministic observation samples (detail in Hersbach, 2000). A smaller CRPS value indicates better probabilistic simulation and the CRPS score of a perfect simulation would equal to 0. Therefore, the changes in overall accuracy of the SD ensemble simulations can be measured by CRPSS. However, unlike the CRPS score, the optimal CRPSS score is equal to 1 and negative values indicate a negative improvement with respect to the reference control run.

## 3. Results and discussion

## 3.1 Open-loop ensemble simulations

In order to investigate the impact of meteorological perturbations on snow simulations, an ensemble containing 100 SD simulations derived from as many different meteorological conditions was analyzed. For the sake of concision and clarity, we considered only one winter season for implementing snow simulation experiment at each site, and the results are shown in Figure 2. As shown in Figure 2, the possible overestimation and underestimation of SD simulations produced by the perturbation forcing data were contained within the ensemble spread, which is a direct consequence of the perturbation of the forcing data. Since the meteorological perturbations are unbiased, the physical processes with nonlinear characteristics within the model is supposed to be the main reason for the uncertainty (Piazzi et al. 2018). During the winter season in northern hemisphere, precipitation and air temperature are primary factors that can determine the total amount of snow.

As Figure 2 shows, the intervals of SD ensemble are significantly different at different sites, although an identical meteorological perturbation method was used. At some sites, such as ATY, MOHE, WFJ, and CDP, larger SD ensemble spreads were obtained, and most of the SD observations were covered by the ensemble spread. In this case, high-quality particles can be directly selected from the ensemble. However, at some other sites, such as ROPA, SDA, and SASP, narrow SD ensemble spreads were obtained, and the uncertainty interval of simulated SD can hardly cover the observations. In this case, the so-called high-quality particles cannot even be found in the ensemble, and the model prior error becomes a prerequisite for successful assimilation at this time. Especially at the ROPA site, the snow cover was extremely unstable, resulting in difficulty in figuring out any variation rules of SD. The narrow SD ensemble spread at this site also demonstrates that precipitation and air temperature were not the main factors causing snow change. According to the literature, sublimation losses at ROPA ranged from 24% to 33% of total annual ablation and occurred 60% of the time during which snow was present. A high sublimation rate may be the main reason for snow instability (Herrero et al., 2016; You et al., 2020a). This directly leads to a perfect ensemble spread that can cover all observations cannot be produced by perturbing the air temperature and precipitation. Generally speaking, the ensemble produced by perturbing air temperature and precipitation does not contain high-quality particles at this site. It was found that the spread of SD ensembles increases when a snowfall event occurs because the perturbation in precipitation would provide different input snow rates for model realization at all sites. Despite this, we still found that the simulated SD deviated significantly from the observation. For example, at SNQ site, the maximum value of simulated SD was almost half the maximum value of observed SD. In this case, it is impossible to obtain a simulated SD ensemble spread that can cover or nearly cover the observation through perturbing the meteorological forcing data. On the one hand, precipitation and air temperature are not the dominant factors affecting snow cover change, which leads to a narrowed ensemble spread at these sites. On the other hand, although the variation trend of snow cover can be accurately expressed by the Noah-

MP model, serious underestimation of the simulated SD shows that the snow simulation performance of Noah-MP is poor at these sites. Nonetheless, the simulated ensembles will be improved whenever the prior error of model state is considered.

## 3.2 DA simulations with perturbed forcing data

Generally, the ability of a model to simulate autonomously can be limited if observation data is assimilated too frequently, resulting in assimilation results that are essentially the same as the observations and do not reflect the differences among models. To address this, the site's SD measurements were assimilated into the Noah-MP model with an observation frequency of five days in this study, enabling the GPF to perform differently at distinct sites. Figure 3 shows the SD assimilation results across snow climates, indicating a substantial improvement in the SD simulations with satisfactory assimilation performance at all sites. The GPF algorithm can handle not only serious underestimations, such as at SNQ, SDA, but also overestimations during the snow ablation period, as seen at CDP, SASP, ATY, and MOHE sites. These results demonstrate the effectiveness of the GPF algorithm as a snow data assimilation scheme and its ability to significantly improve SD simulations, despite the numerous overestimations and underestimations that may occur in the Noah-MP model's snow simulation results across snow climates.

The effectiveness of GPF in updating SD simulations is demonstrated by the KGE values of the DA simulations with perturbed meteorological forcing data, as shown in Figure 4. Although the mean ensemble simulations of SD exhibit substantial improvement at all sites, not all ensemble members were improved, as per the distribution of GPF-DA KGE values. Some ensemble members achieved significant improvement at sites like SDA, SASP, MOHE, and SNQ, while others showed only slight improvement at sites like ATY, WFJ. Figure 4 also reveals that updating SD model simulations at ROPA and WFJ sites is more challenging. Snow simulation performance at the ROPA site is known to be poor due to the high sublimation rate. Certainly, the median value of SD ensemble prediction KGE values is expected to be below zero at this site, indicating that there are few qualified simulations in the prediction ensemble. While the GPF succeeds in enhancing the SD simulations at ROPA, the distribution of GPF-DA KGE values is not concentrated enough, with the 25th percentile approximately at 0.2 and the 75th percentile at about 0.7, indicating that the GPF assimilation algorithm cannot enhance all members but can raise the mean level and obtain an approximation of the optimal posterior estimation. Conversely, the assimilation of snow measurements at CDP site resulted in poor quality of the SD simulations compared to the open-loop ensemble simulations. The median value of GPF-DA KGE was lower than the median value of OL KGE, indicating that a considerable number of ensemble simulations failed to capture the observed values after assimilating snow measurements. However, Figure 3 shows that the mean ensemble simulations after assimilating snow measurements are much closer to SD observations. Thus, it underscores the importance of the ensemble mean in characterizing the filter effectiveness and the approximate value of the optimal

posterior estimation of model state. Additionally, the scale of the model ensemble spread was found to be the determinant factor that significantly affects assimilation results. A large ensemble spread can adjust the simulations toward the observed system state even if the model predictions are heavily biased.

Figure 5 displays the CRPSS value of GPF-DA at different sites. The smaller the CRPSS value, the worse the probabilistic simulation (with an optimal score of 1). The highest CRPSS score of 0.91 was achieved at SASP, while the lowest score of 0.44 was observed at CDP. These results indicate that the GPF enhances the overall accuracy of ensemble simulations most at SASP and least at CDP with respect to the open-loop ensemble simulation. Certainly, this cannot be illustrated by the mean ensemble simulations (Figure 3) but is consistent with the KGE statistical results (Figure 4). Although the open-loop simulations at SNQ exhibited serious underestimation, a satisfactory assimilation result was obtained at this site with a CRPSS score of 0.87. At the SNQ site, the snow simulation performance of Noah-MP model is poor and the model shows serious underestimation during snow stable phase. Implementing a data assimilation experiment in this case is a tricky business since it is difficult to obtain a suitable simulated ensemble by perturbing the meteorological forcings. However, since the model prior error was considered in GPF algorithm, the overall accuracy of the ensemble simulations will be substantially enhanced and this is the reason why a satisfactory assimilation result at SNQ site can be obtained. ROPA was found to be a difficult site to enhance the overall accuracy of ensemble simulations, with a CRPSS score of only 0.58. The snow cover was highly unstable, and the variation of SD exhibited extreme irregularity, which may be the main obstacles to snow data assimilation at this site.

Based on these findings, we conclude that the effectiveness of GPF varied among snow climates: it can be employed as a snow data assimilation scheme across snow climates, however, its performance varied across different sites. It is necessary to explore the sensitivity of measurement frequency and ensemble size for the GPF assimilation scheme at various sites.

## 3.3 Sensitivity analysis of DA scheme to SD measurement frequency

For complex land/snow process models, model errors can gradually lead to the system deviating from the true value. Therefore, it is necessary to continuously incorporate observations into the model framework to adjust the operating trajectory of the state. Obviously, the frequency of incorporating observations, that is, the assimilation interval, has an important impact on the assimilation system. To investigate the effect of the SD measurement frequency on the performance of GPF, we conducted a sensitivity experiment at eight sites. We aimed to determine how reducing the frequency of SD measurements affects the DA simulations. As expected, a decrease in SD measurement frequency led to a reduction in the impact of the GPF updating on the model simulations, resulting in a gradual increase in the mean RMSE value. Figure 6 illustrates the RMSE ensembles of SD simulations resulting from assimilating different frequency SD measurements over the snow period at each site.

Higher frequency SD assimilation improves the accuracy of the simulated SD, as shown by the lower RMSE value achieved when the frequency of SD measurement was set to five days. This means that more frequent SD measurements improve the accuracy of the model, which is particularly useful in regions where snow conditions can change rapidly. The range of RMSE values at different sites varied significantly, as it was related to the maximum value of SD. For instance, a thick snow at SNQ and WFJ sites during the snow period led to larger RMSEs of SD simulations. Notably, an increase in the length of the assimilation window generally resulted in a significant increase in the RMSE value. However, an abnormal occurrence was observed at the SDA site, where the assimilation effect of 20 days of SD measurements was significantly better than that of 15 days. Although the RMSE distribution of SD assimilation results with 20 days of observations appeared superior to that of 15 days, the RMSE mean values of the two were very close: 0.08 m and 0.07 m, respectively. Therefore, this anomaly can be ignored. These results indicate that the frequency of SD observations has a significant impact on the effectiveness of the GPF algorithm and that a dense amount of observational data can effectively improve the assimilation results.

## 3.4 Sensitivity analysis of DA scheme to ensemble size

The results of the experiment aimed at evaluating the impact of particle number on the assimilation performance of GPF are presented in Figure 7. As expected, increasing the particle number up to the threshold leads to a significant improvement in the percent effective sample size. However, the filter performance does not improve significantly when the particle number exceeds the threshold. Figure 7 shows that the GPF algorithm yields the minimum error at all sites when the particle number is set to 100, indicating that one hundred particles can optimize the performance of the GPF algorithm. Although a large particle number can enhance particle diversity and prevent filter divergence, it increases the computation burden without reducing the system error. As illustrated in Figure 7, the RMSEs are generally at the same level when the particle number equals 120 and 160, and they are significantly larger than the RMSE when the particle number is equal to 100. The slight impact of the change in the particle number on the performance of GPF, when the particle number is below the threshold, indicates low system sensitivity to the ensemble size, and this is observed at all sites. Essentially, blindly increasing the particle number does not guarantee a better DA performance of the GPF algorithm. As demonstrated in Figure 7, the RMSEs of simulated snow-depth are virtually unchanged at all sites, despite an increase in the particle number from 120 to 160. This suggests that blindly increasing the ensemble size only increases the computational burden without improving the performance of the GPF.

## 3.5 Compared to traditional resampling methods

To demonstrate the effectiveness of using genetic algorithms for particle resampling, we compared the results of our genetic algorithm (PF-G) to those of traditional resampling methods:

systematic resampling (PF-S) and multinomial resampling (PF-M), which are both commonly used in particle resampling. The calculation process for these methods is detailed in the particle filter introduction references. Figure 8 shows the RMSE values for SD simulations obtained using these three methods. We found that the PF-G outperforms PF-M and PF-S at all sites, as evidenced by the significantly smaller mean and median RMSE values. This indicates that the PF-G is suitable for snow data assimilation in various snow climates and is somewhat superior to traditional particle filters. At most sites (MOHE, ATY, SDA, and ROPA), PF-M and PF-S showed similar performance, meaning that these methods did not produce a significant difference in the assimilation results. This is because these traditional resampling methods can only mitigate particle degeneration by resampling particles, but are unable to prevent particle impoverishment. Therefore, they are unable to select high-quality particles and keep the particles have variety. Significantly, the mean and median RMSE values for PF-G were lower than those of PF-M and PF-S at several sites (SASP, SNQ, and WFJ) where the snow cover was relatively thick, with maximum SD during the snow period reaching 2.45 m, 2.95 m, and 2.40 m, respectively. This suggests that PF-G performs better in assimilating data from thick snow covers.

The multinomial and systematic resampling methods select particles from the original particle set at different levels or based on the accumulation of particle weights. Both of the resampling methods extract particles from the entire particle set, and the corresponding particle values do not undergo any essential changes. However, when compared to the two traditional particle resampling methods, the genetic algorithm first uses the fitness function to calculate the "survival rate" of each particle one by one, and then performs crossover, mutation and other operations on the selected particles. This approach ensures that the resampled particles are high-quality particles, which is the main reason why genetic particle filtering has an advantage in the snow data assimilation experiments. As Figure 8 shows, the assimilation error of the genetic particle filter is the smallest at all sites. From the results of the real assimilation experiment, it can be seen that genetic particle filtering has more advantages over the other two methods.

# 4. Conclusions

In this study, we investigated the potential of using GPF as a snow data assimilation scheme across eight sites with varying snow climates. We addressed the problem of degeneration and impoverishment in PF algorithm by using the genetic algorithm to resample particles. We also examined the sensitivity of GPF scheme to measurement frequency and ensemble size. The main findings of this study are as follows:

1. The GPF was an effective snow data assimilation scheme and can be used across different snow climates. The genetic algorithm effectively addressed the problem of particle degeneration and

impoverishment in the PF algorithm.

2. Our experiment showed that the system has low sensitivity to the particle number, and 100 particles can achieve a better assimilation result across different snow climates. This indicates that 100 particles are suitable for representing the high dimensionality of the system.

3. We found that perturbations in meteorological forcing data were not sufficient to provide ensemble spread, resulting in poor filter performance. Particle inflation can make up for this deficiency. Moreover, we observed that the RMSE of simulated SD decreased significantly with the increase of the frequency of SD measurement, indicating that dense observational data can improve the assimilation results.

4. Compared to the two classic resampling methods, the particle filter with genetic algorithm as resampling method shows a better assimilation performance especially in a thick snow cover, the distributed RMSEs are more centralized and a smaller mean error will be obtained.

Our experiments were based on forcing data and snow observations from various sites with different snow climates. While our results provide a reference for applying GPF to snow data assimilation, further research is needed to investigate the performance of GPF on a regional scale and to explore the assimilation of snow observational data from remote sensing or wireless sensor networks into land surface models using GPF. In summary, our study demonstrates the feasibility of using GPF for snow data assimilation and provides valuable insights for future research in this area.

# Acknowledgements

Our research received support from several sources, including the National Natural Science Foundation of China (grant number 42101361, 42130113, 41871251, and 41971326), the Scientific research project of higher education institutions in Anhui province, and the Key Research and Development Program of Anhui Province (2022107020028).

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

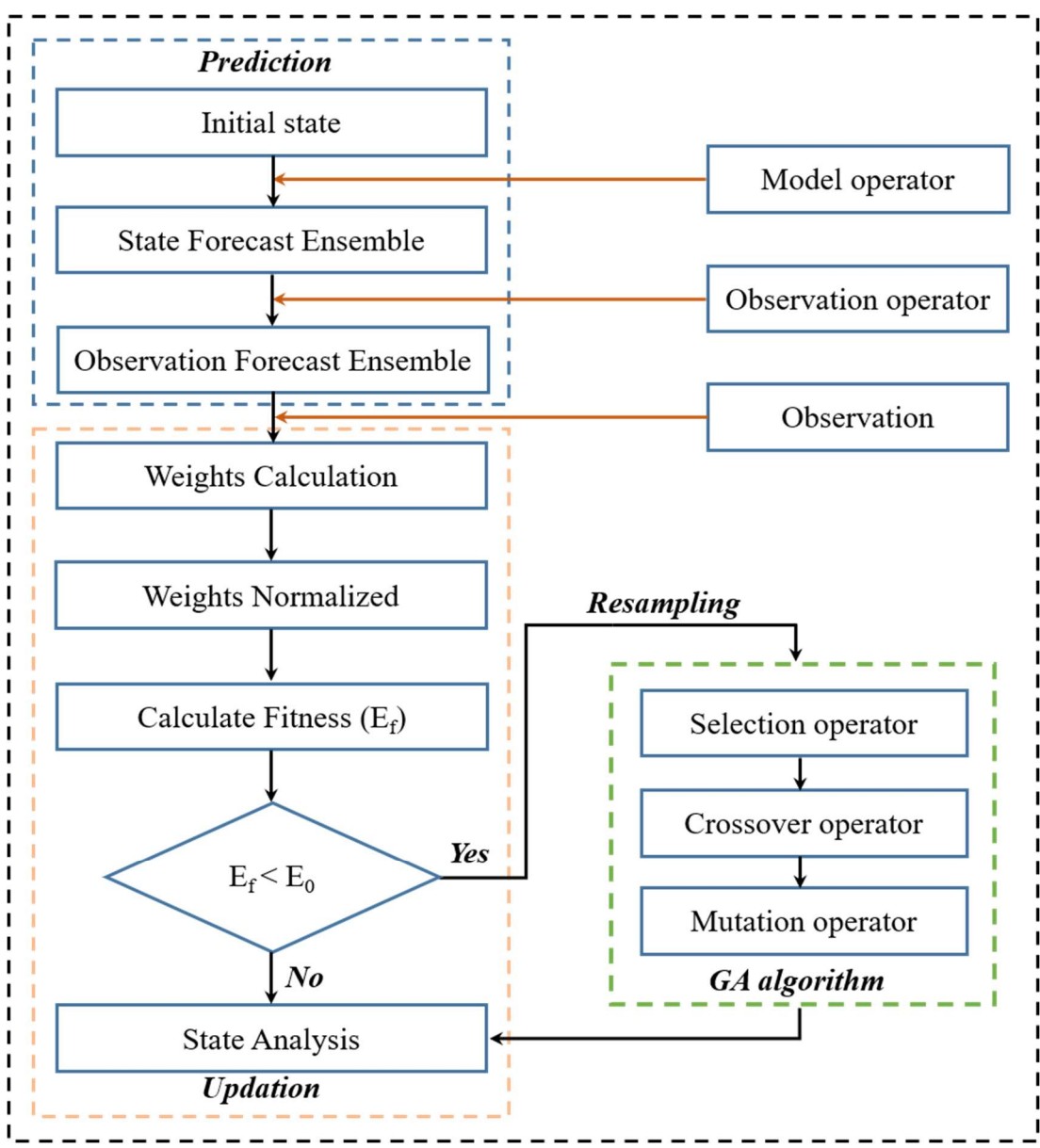

**Figure 1.** Flowchart of Genetic particle filter

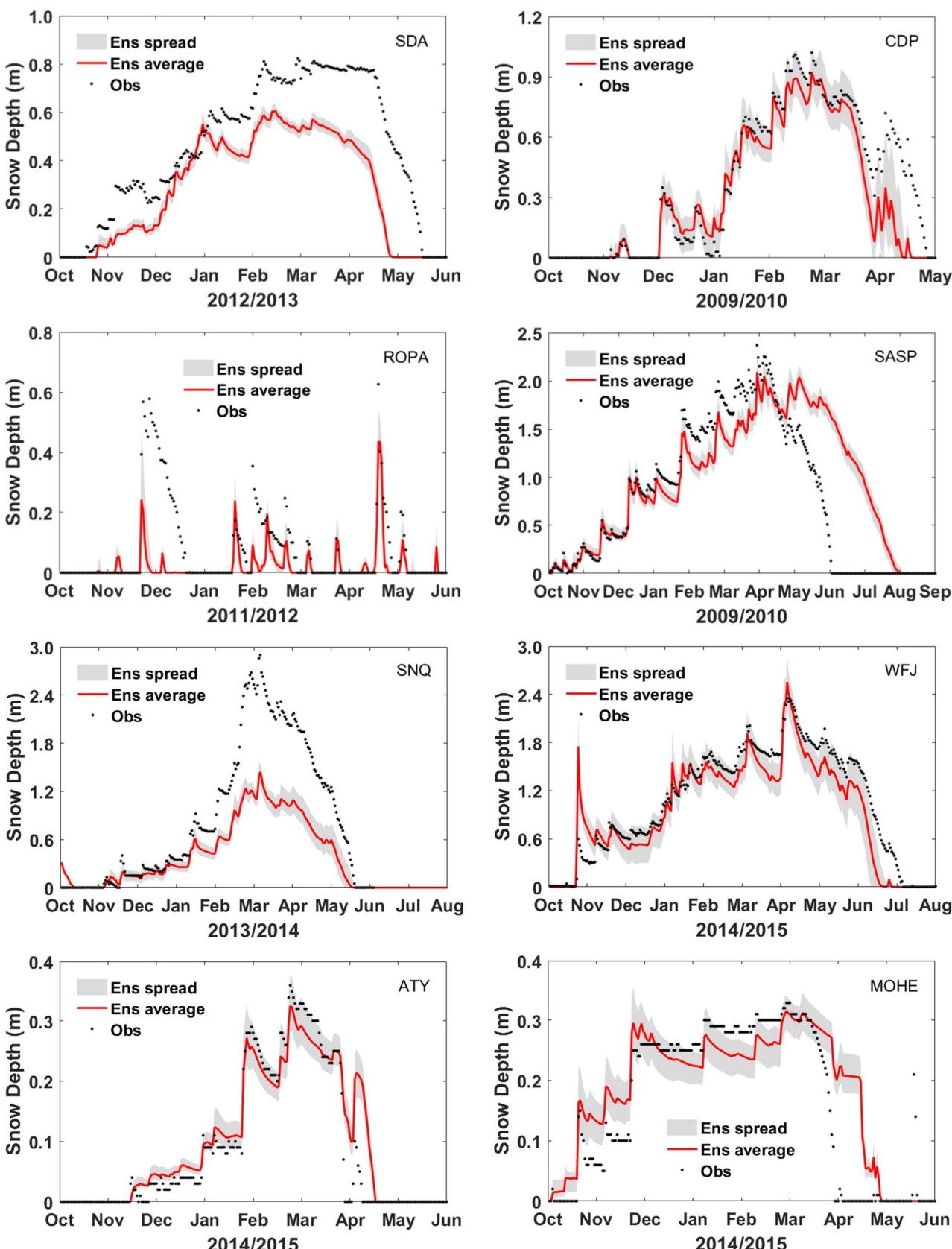

**Figure 2.** Impact of the meteorological uncertainty on snow depth ensemble simulations

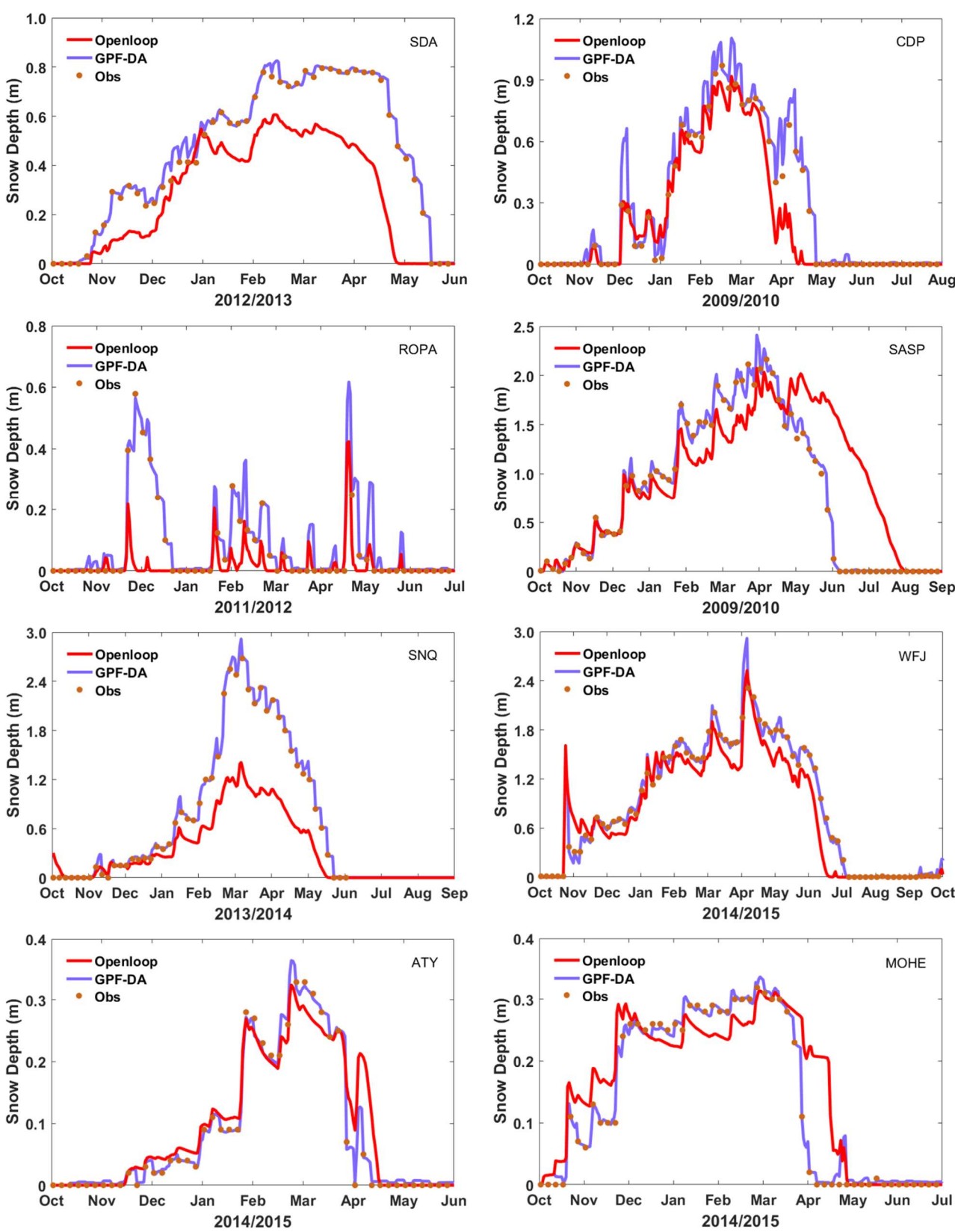

**Figure 3.** Evaluation of the SD at eight sites from mean ensemble simulation and assimilation with

the measurements.

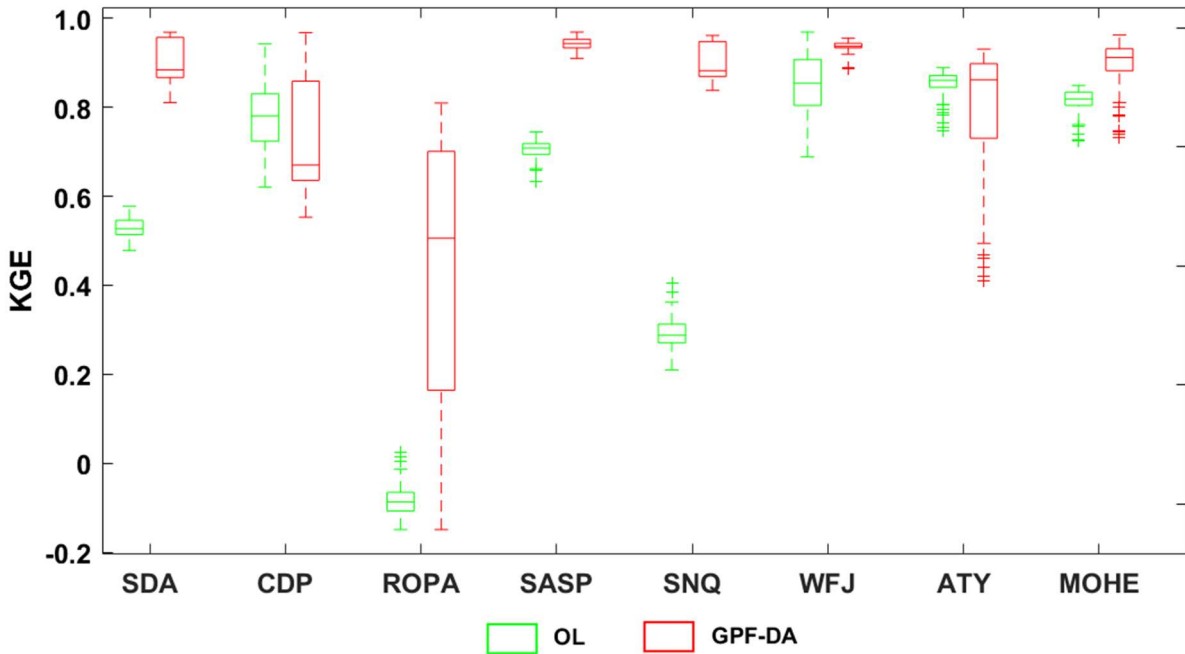

**Figure 4.** The KGE values of SD simulations, the OL and GPF-DA are in green, red, respectively.

The bottom and top edges of each box indicate the 25th 75th percentiles, respectively. The line in the

middle of each box is the median.

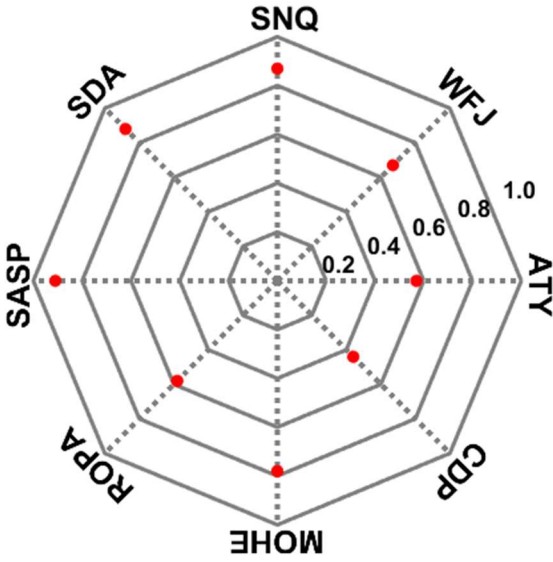

**Figure 5.** Comparison of the CRPSS value of GPF-DA at different sites.

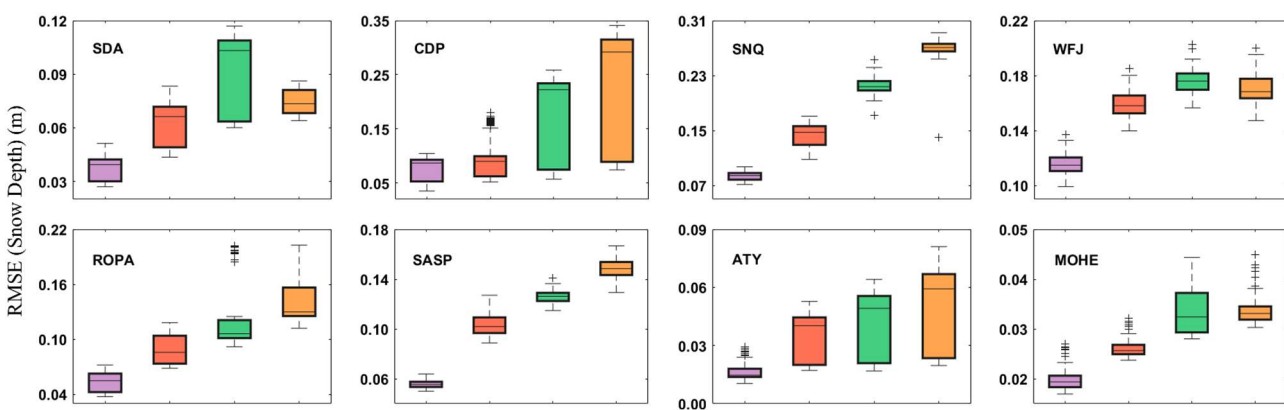

**Figure 6.** The RMSE values of SD simulations at different sites, from left to right in each subfigure

are the assimilation observation frequency is 5, 10, 15, 20 days, respectively, and with different colors.

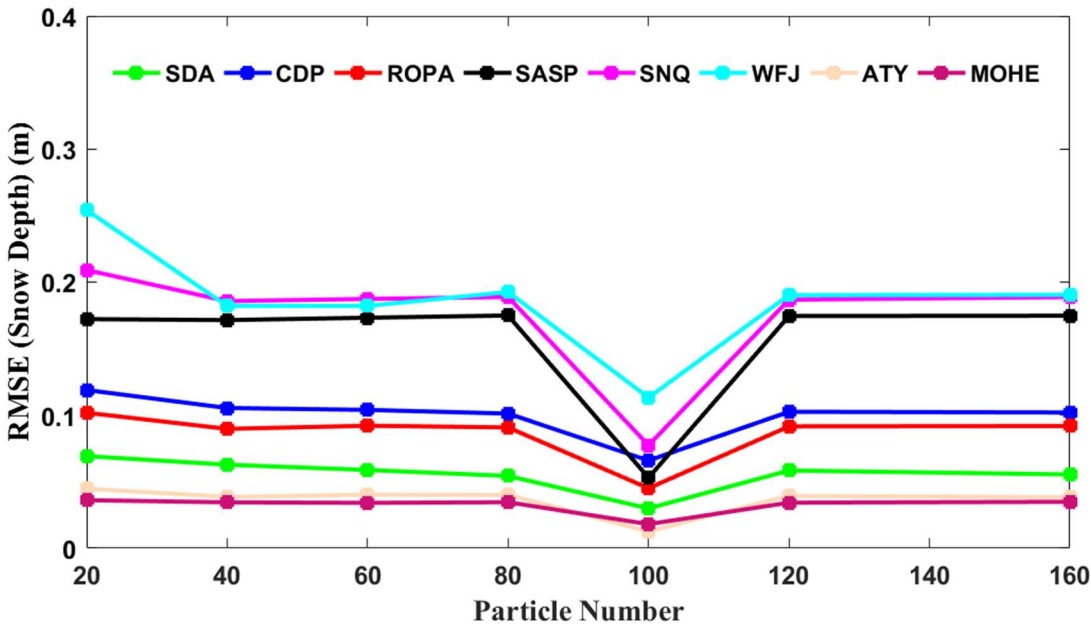

**Figure 7.** Sensitivity analysis of the GPF snow DA scheme to particle number at eight sites, during

different snow periods.

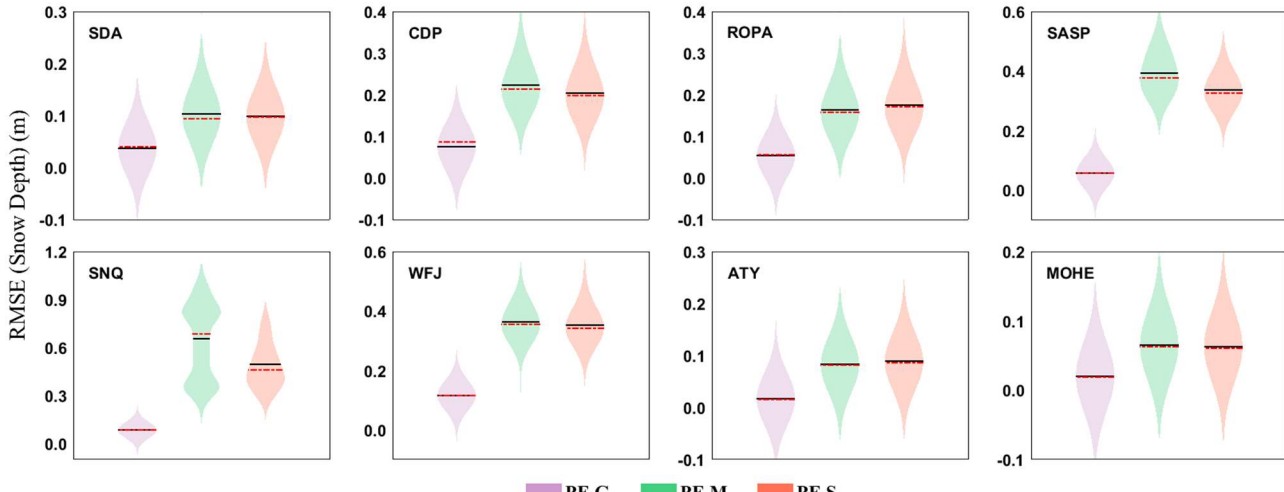

**Figure 8.** The RMSE values of SD simulations by three different resampling methods. For each subfigure, from left to right are the particles resampled by genetic algorithm, multinominal method, systematic method, respectively, and with different colors, the black line indicates the mean, and the red line indicates the median; the kernel bandwidth was 0.05.