# Peer review of "A genetic particle filter scheme for univariate data assimilation into Noah-MP model across snow climates"

_Hydrology and Earth System Sciences, 2022_

## Author Comment (AC1)

**Response to comments by Reviewer #1:**

We would like to take this opportunity to gratefully thank the reviewer for his/her constructive comments and recommendations for improving the paper. An item-by-item, point-by-point response to the interesting comments raised by the reviewer follows.

The topic addressed is within the scope of HESS. The manuscript is generally well organized and results are clearly presented. This manuscript investigated the potential of GPF used as a snow data assimilation scheme across different snow climates, the results presented in this manuscript will help develop new data assimilation scheme and improve the simulation accuracy of land surface model that leads to improve weather and climate prediction. In my opinion, this manuscript could be accepted for publication in HESS after the following comments are addressed.

Comments:

**Line 106: "Above studies" may need some recent references.**
**Reply:** We agree that recent literature on particle filter is missing in the Introduction section, and we will include recent literature in this section.

**Line 245: "The number of particles was set to 100" have been expressed in line 225, I suggest deleting one.**
**Reply:** Thanks for your sincere and constructive suggestions and we will delete this sentence.
**Line 250: the variance scaling factor of the temperature was set to 2.0, why this value was chosen, 3.0 or other value can be used here?**
**Reply:** Thanks for your sincere and constructive suggestions. The variance scaling factor of the temperature was referenced the method of Lei et al. (2014), and the value was obtained through repeated attempts and experiments.

**Line 259: What does the "SD" is refer to? Do you mean SD is the abbreviation of snow depth?**
**Reply:** Thanks for your sincere and constructive suggestions. Here the SD is the abbreviation of snow depth and we will remark in the text.

**The abstract should provide some numerical values from the performance metrics of the results.**
**Reply:** Thanks for your sincere and constructive suggestions. We will provide some numerical values from the performance metrics of the results in the abstract and rewrite the abstract seriously.

**Line 236: Except for the air temperature and precipitation can be perturbed, whether other meteorological forcing variable can be perturbed, such as relative**

**humidity and wind speed? As far as I know, the wind speed has great impact on the distribution of snow.**

**Reply:** Thanks for your sincere and constructive suggestions. The snow depth is influenced by precipitation and temperature, specially, snow is highly sensitive to the air temperature. We agree that the wind speed and other meteorological forcing variables, like longwave radiation, shortwave radiation, have great impact on the distribution of snow. In this study, we attempt to examine the performance of genetic particle filter at site scale and the distribution of snow at regional scale was not considered, so we just perturbed the air temperature and precipitation.

**The English writing has to be polished.**

**Reply:** Thanks for your sincere and constructive suggestions. We will proof-read the whole manuscript to fix the language and grammar issues, and an English writing service will be purchased for this paper.

---

## Author Comment (AC4)

This paper optimizes the snow depth data simulated by the model by using the snow depth data observed at the station. The main questions are as follows:

1. The title of the article is not accurate, and the purpose of assimilation cannot be obtained. The meaning of "snow" is too broad, so it needs to be specific;

**Reply:** Thanks for your sincere and constructive suggestions. In this study, the Genetic Particle Filter was used as a data assimilation scheme to improve the simulation accuracy of snow depth by Noah-MP model, as the reviewer said, the meaning of "snow" is too broad in the title, and we will change a new title which can highlight the theme of this article.

2. Is the Ws in NOAH-MP optimized by snow depth observation? Ws is the snow water equivalent. Is there any difference between the two?

**Reply:** Thanks for your sincere and constructive suggestions. The snow water equivalent was calculated by snow depth and snow density in Noah-MP model, and the snow water equivalent was optimized by snow depth observation. In this study, most of the sites do not have snow water equivalent observation data, so the experimental results of snow water equivalent were not shown in the paper.

3. The observation operator and model operator mentioned in Flow Chart 1 are not seen in the text, and need to be clarified;

**Reply:** Thanks for your sincere and constructive suggestions. In this study, we just assimilated the station observation data into the Noah-MP model, and we haven't assimilated remote sensing observation data into the model, the observation operator is an identity matrix in this study. And the model operator is the Noah-MP model.

4. The optimized state variables are not seen in the paper. If only the simulation results of NOAH-MP are corrected, which can not be regarded as assimilation;

**Reply:** Thanks for your sincere and constructive suggestions. In this study, the Genetic Particle Filter (GPF) was used to improve the simulation accuracy of snow depth by Noah-MP model, and the performance of GPF was investigated across snow climates. We don't understand what you mean, can you be more specific?

5. The calculation formula of fsnow, g needs to be given. Snow depth is also used in the calculation formula of fsnow, g. If snow depth is assimilated, has this been considered? If regional assimilation is carried out, how is fsnow, g calculated?

**Reply:** Thanks for your sincere and constructive suggestions. The main work of this study is to investigate the performance of Genetic Particle Filter across snow climates, and examine the feasibility of GPF used as a snow data assimilation scheme in point-scale. If snow depth is assimilated, the fsnow can also be updated. The regional assimilation is our next work, the snow albedo will be assimilated into land surface model to improve the fsnow. Here, we verified the feasibility of GPF used as snow data assimilation scheme and prepare for the regional assimilation experiment.

6. The introduction of assimilation process in this paper is not complete and detailed enough, and needs to be further improved.

**Reply:** Thanks for your sincere and constructive suggestions. We will thorough revise the manuscript according to the reviewer's suggestion.

---

## Author Comment (AC5)

Review on the "Investigating the performance of Genetic Particle Filter in snow data assimilation across snow climates" by You et al.

Snow is a critical hydrothermal variable in land surface that influences the surface energy and hydrological cycles. Snow data assimilation is also important to improve the modeling accuracy of snow and thus facilitates other related processes (e.g., albedo, snow melting runoff, and temperature). You et al. investigated the performance of genetic particle filter in snow data assimilation over eight stations and discussed the influences of different assimilating frequency and particle number. Although I am not very familiar with the data assimilation method especially the mathematical rule, I think the current research needs at least major revision to illustrate its novelty, introduce the method more clearly, analyzing the results in depth, and proofread the manuscript carefully. Detailed comments are below:

Major comments:

The novelty of this research is not clear. If the genetic particle filter is a new method, then its advantages against other methods should be investigated directly. If the "across snow climates" is a novelty, then I do not think the eight stations can represent snow climates considering the highly heterogeneous snow distribution. Finally, "the higher assimilating frequency, the higher simulation performance" is not surprising. Thus, I suggest the authors to clarify the novelty more clearly, so as to help the reader get the importance of this work. The method needs further introduction. Although the genetic particle filter data assimilation scheme is introduced in section 2.3, I am still confused that which variable you assimilate into the land surface model? If you only assimilate the snow depth, then how do you deal with other snow variables (e.g., snow water equivalent, snow density and snow age)? Also, how do you deal with the potential inconsistency between snow and ground temperature (for example, when the model shows no snow and the ground temperature is above zero, then how to assimilate the observed snow depth)? Actually, I am very concerned about the assimilation and evaluation. In figure 3, it seems you assimilate the observation every 5 days and then evaluate the model simulation at the same step? If this is the case, then will the direct insertion method show higher performance than the genetic particle filter?

**Reply:** Thanks for your sincere and constructive suggestions. Here, we assimilated the snow depth observation every 5 days, the temporal resolution of the simulation result and observation of snow depth are one day. To be honest, we have used the direct insertion method to improve the simulation result, however, the performance of direct insertion method even worse than particle filter.

The spatial difference. It seems the spatial difference among different stations is not strong and few information can be get (except the robust of the result, may be). Some insightful analysis on the spatial difference may help improve the manuscript. The writing needs careful proofreading.

**Reply:** Thanks for your sincere and constructive suggestions. We will thorough revise the manuscript according to the reviewer's suggestion, thank you very much.

For example:

L48: "succeeds in catching snow dynamics is" may be "succeed in catching snow dynamics is"

**Reply:** Thanks for your sincere and constructive suggestions. We have revised in the manuscript.

L51: "is aimed at investigating ... and obtain the ..." may be "is aimed at investigating ... and obtaining the ..."

**Reply:** Thanks for your sincere and constructive suggestions. We have revised in the manuscript.

L60: "However, this method possible result in ..." may be "However, this method possible results in ...".

**Reply:** Thanks for your sincere and constructive suggestions. We have revised in the manuscript.

L68: "this method does not require a model a model linearization." what do you mean?

**Reply:** Thanks for your sincere and constructive suggestions. The Kalman Filter is a useful tool for linear system but it requires a model linearization for a nonlinear system, the Monte Carlo approach was employed to approximate error estimates based on an ensemble of model simulations in Ensemble Kalman Filter (EnKF). As a result of this, the EnKF does not require a model linearization when used in nonlinear system.

---

## Author Comment (AC6)

**Response to comments by Editor:**

Dear authors,

This manuscript uses field data to improve snow modelling through the genetic particle filter algorithm. The topic is highly relevant and fits nicely within this SI. However, the manuscript requires significant improvements in order to be considered for possible publication in HESS. The Introduction section should be shorted, more concise and better highlight the research question and the need for this study, and use more updated references. The study sites must be better described, namely regarding the snow differences between the sites, so that we can understand the real application of your proposed method. The methodology requires relevant improvements to describe the genetic particle filter algorithm, and all the mathematical assumptions performed. The Results must be clearly presented and discussed. Discussion should clearly present the advantages of the proposed method comparing with others, and the limitations linked with the assumption performed. It is also important to compare the results with those from previous studies, and bring more references into this section. The Conclusions section must clearly identify the novelty and main messages of this study, and clearly identify why the proposed method is better than the available ones. Language editing is also required. More detailed comments have been provided by reviewers and must be considered in improving the manuscript. There was one late reviewer whose comments are provided bellow and should be also considered.

**Reply:** Thanks for your sincere and constructive suggestions. As you said, some significant weaknesses in this manuscript and we will thorough revise the manuscript according to the Editor and Reviewer's suggestion, thank you very much.

Reviewer 6:

In this manuscript, You et al. set up a particle filtering framework using the Genetic Algorithm to avoid particle filter-typical degeneracy and sample impoverishment issues. They apply this framework to snow depth measuring sites in different climatological regions, expecting to learn about particle filter performance at each of the sites. They analyze the assimilated snow depth with respect to the suitability of their particle filtering algorithm for application in different snow climates, the influence of the used particle number on performance metrics, and the influence of assimilation window length on the performance. The manuscript is structurally well-organized. The topic is in general very interesting and the effort to push the field of data assimilation forward is very much appreciated. However, in my opinion there are some significant weaknesses in this manuscript:

- a lacking motivation of the research question(s') relevance

**Reply:** Thanks for your sincere and constructive suggestions. The original goal of this study is to investigate the performance of genetic particle filter used as a snow data assimilation scheme across snow climates and we attempted to resample the particles using genetic algorithm, we will thorough revise the manuscript according to your suggestions.

- a superficial description of the measurement sites and their properties, making a meaningful interpretation of the results with respect to the research question(s) difficult.
**Reply:** Thanks for your sincere and constructive suggestions. we will thorough analyze the experimental results according to the reviewer's suggestion, thank you very much.

- a lacking presentation of the Genetic Algorithm and stressing why this method is the most suitable for the analysis
**Reply:** Thanks for your sincere and constructive suggestions. The degeneracy and sample impoverishment issues are common faults of particle filter, here, the genetic algorithm was used in resampling process was expected to effectively overcome these issues.

- a superficial interpretation of the results, in particular with respect to the overarching hypothesis (different filter performance in different "snow climates")
**Reply:** Thanks for your sincere and constructive suggestions. We will thorough revise the manuscript according to your suggestions.

- issues with the used literature in the References section and in general a rather scarce literature selection. Difficulties in the use of English, which makes some sections of the manuscript hard to understand
**Reply:** Thanks for your sincere and constructive suggestions. We will thorough revise the manuscript according to your suggestions.

- an intransparent (or simply not listed?) choice of model parameter values and meteorological values to perturb; unclear or not explained error distribution choices.
**Reply:** Thanks for your sincere and constructive suggestions. We will introduce the meteorological perturbation method in the manuscript.

- a results and discussion section that partly loses contact with the research questions
**Reply:** Thanks for your sincere and constructive suggestions. We will thorough revise the manuscript according to your suggestions.

If this manuscript is accepted for a major revision process, it should be largely rewritten and then undergo line-by-line comments in a second review iteration. The focus should first be on the following aspects:
- reworking the manuscript research questions (is it about the filter performance in different climates as the title suggests or about the three questions formulated at the end of the introduction, or both?)
**Reply:** Thanks for your sincere and constructive suggestions. The original goal of this study is to investigate the performance of genetic particle filter used as a snow data assimilation scheme across snow climates

- a more comprehensive literature review on the technical literature regarding the research questions

**Reply:** Thanks for your sincere and constructive suggestions. We will thorough revise the manuscript according to your suggestions.

- a more detailed description of the used particle filter method and why this filter is chosen to be the most suitable to answer the research question
**Reply:** Thanks for your sincere and constructive suggestions. We will thorough revise the manuscript according to your suggestions.

- a more critical questioning of the results, in particular with respect to the 100-particle threshold (e.g. why in Fig. 7 a minimum exists at 100 particles).
**Reply:** Thanks for your sincere and constructive suggestions. We will explain the reasons in the revised manuscript.

---

## Author Response (AR1)

**Response to Comments on the Manuscript:**

**"Investigating the performance of Genetic Particle Filter in snow data assimilation across snow climates"**

**(HESS-2022-350)**

March 26, 2023
* * *
The authors gratefully acknowledge the editors and the anonymous reviewers for their constructive comments. We have made a comprehensive revision of our previous manuscript. The main modifications have been highlighted in yellow color in order to highlight the issues raised by the editors and anonymous reviewers, and are summarized as follows:

For more details, please refer to the item-by-item response. Thank you for your time.

**Response to comments by Reviewer #1:**

We would like to take this opportunity to gratefully thank the reviewer for his/her constructive comments and recommendations for improving the paper. An item-by-item, point-by-point response to the interesting comments raised by the reviewer follows.

The topic addressed is within the scope of HESS. The manuscript is generally well organized and results are clearly presented. This manuscript investigated the potential of GPF used as a snow data assimilation scheme across different snow climates, the results presented in this manuscript will help develop new data assimilation scheme and improve the simulation accuracy of land surface model that leads to improve weather and climate prediction. In my opinion, this manuscript could be accepted for publication in HESS after the following comments are addressed.

Comments:

**Line 106: "Above studies" may need some recent references.**
**Reply:** Thanks for your sincere and constructive suggestions. We have included recent literature in this section.

**Line 245: "The number of particles was set to 100" have been expressed in line 225, I suggest deleting one.**
**Reply:** Thanks for your sincere and constructive suggestions. We have deleted the sentence in this line.

**Line 250: the variance scaling factor of the temperature was set to 2.0, why this value was chosen, 3.0 or other value can be used here?**

**Reply:** Thanks for your sincere and constructive suggestions. The variance scaling factor of the temperature was referenced the method of Lei et al. (2014), and the value was obtained through repeated attempts and experiments.

**Line 259: What does the "SD" is refer to? Do you mean SD is the abbreviation of snow depth?**

**Reply:** Thanks for your sincere and constructive suggestions. Here the SD is the abbreviation of snow depth and we have remarked in the manuscript.

**The abstract should provide some numerical values from the performance metrics of the results.**

**Reply:** Thanks for your sincere and constructive suggestions. We have provided some numerical values from the performance metrics of the results in the abstract and rewrite the abstract seriously.

**Line 236: Except for the air temperature and precipitation can be perturbed, whether other meteorological forcing variable can be perturbed, such as relative humidity and wind speed? As far as I know, the wind speed has great impact on the distribution of snow.**

**Reply:** Thank you for your sincere and constructive suggestions. Snow depth is primarily influenced by precipitation and temperature, with snow being highly sensitive to air temperature. Additionally, wind speed and other meteorological forcing variables such as longwave radiation and shortwave radiation have a significant impact on the distribution of snow. Although our study focuses on the performance of the genetic particle filter at the site scale, we acknowledge that the distribution of snow at the regional scale was not taken into consideration. Therefore, we only perturbed the air temperature and precipitation.

**The English writing has to be polished.**

**Reply:** Thanks for your sincere and constructive suggestions. We have proof-read the whole manuscript to fix the language and grammar issues, and an English writing service has been purchased for this manuscript.

**Response to comments by Reviewer #2:**

Snow depth simulation is difficult in land surface models and the data assimilation for snow depth is of importance for cold regions hydrology and energy balance modeling. This manuscript tried to propose a new data assimilation scheme for the land surface model. I read it with high interest but did not find out the logic of this manuscript. Therefore, I cannot give too positive evaluation at the current stage, I would suggest a thorough revision before it can be considered for publication in the journals. first, the language is quite poor and the writing is difficult to understand especially in the introduction and results and discussion parts. I can't understand the importance of this work except from my own understanding of the cold regions modeling and data assimilation. Second, some of the references are not shown in the references part even they are put in the main text. This is awful and I feel that the preparation for this work was not serious and also not strictly following the journal's rules. Besides, the results and discussion are quite awful in writing, as I can't find out the useful information from this work concluded by the authors. This is a quite pity issue even the meaningful work was conducted. Given the above mentioned issues, I feel that the detailed comments are not necessary if the authors don't make a thorough revision on the whole story telling logic. Therefore, I suggest a rejection this turn and a chance for resubmission with a clear outline that focuses on the most interesting part of the

work would be a good suggestion from my side. Sorry for being not too positive this time given the current version of the manuscript.

**Reply:** Thanks for your sincere and constructive suggestions. The reviewer has pointed out the deficiency of this manuscript, and we sincerely agree the comments. We do our best to revise this manuscript and examine the language seriously, and we also re-considered the structure of this manuscript. An English writing service has been purchased for this manuscript, and we look forward to your positive comments this time.

**Response to comments by Reviewer #3:**

Summary: The study is focused on the simulation of snow depth with a simple energy balance model with assimilation with observations. The data assimilation scheme is based on a genetic particle filter algorithm. The main conclusion is that this algorithm performs well for snow depth simulations.

Recommendation: the manuscript is in general terms poorly written. The English needs would need extensive copy-editing, and this deficiency often hinders the understanding of technical and scientific aspects of the study. In addition, the manuscript does not include important information that is critical to understand what has been done. One is that there is no description of the genetic particle filter algorithm itself, which is surprising. Unfortunately, I cannot recommend the publication of the manuscript. A revision would entail rewriting the manuscript almost completely.

Main points

1) **The presentation is poor. The English needs very extensive revisions; acronyms are not defined (for instance SD, which I interpret to be snow depth! SWE and GPF are not defined either). The discussion is restricted to the own results and does not place the results in the framework of previous studies (what has been learned, what is the novelty?)**

**Reply:** Thanks for your sincere and constructive suggestions. We have thorough revised the manuscript according to the reviewer's suggestion.

2) **The destitution of important technical aspects is missing. The genetic algorithm is essentially not described the paragraph starting in line 218 is so obscure that essentially nothing can be understood. The reader is not informed of many technical aspects. What are the 'particles'? how are they genetically generated? what are the crossover and mutation operators? why the genetic algorithm would improve on the deficiencies of other particle filters?**

**Reply:** Thanks for your sincere and constructive suggestions. We have introduced the Genetic Particle Filter algorithm in detail in the revised manuscript, including the selection, crossover and mutation process for the particles. The random samples propagating in state space are used to approximate the probability density function of state variable, in this case, the integral operation was replaced with the sample mean to obtain the minimum variance estimate of the state. And the sample members are called "particles" in particle filter. The particles were generated by forcing the model operator with the perturbed meteorological forcing data. The crossover operators are below equations:

$$x_m^{'} = \alpha x_m + (1-\beta) x_n \qquad (1)$$

$$x_n^{'} = \beta x_n + (1-\alpha) x_m \qquad (2)$$

Where $\alpha$ and $\beta$ are the empirical crossover coefficient, $\alpha = 0.45$, $\beta = 0.55$ in this paper. The mutation operator is,

$$x_k^{'} = x_k + \eta * Uniform \qquad (3)$$

where the $Uniform$ represents random of uniform distribution and $\eta$ is the empirical coefficient which was set to 0.01 in this paper. The problem of particle degradation solved by conventional resampling methods like multinominal resampling and systematic resampling always results in particle impoverishment, the diversity of particles will be greatly enhanced using the genetic algorithm in particle resampling. We have revised the manuscript seriously according to the reviewer's suggestion.

3) **I kept wondering of the utility and meaning of some of the mathematical assumptions. For instance, equation 9 seems to be unnecessary complicated. The distribution of the random noise w is just uniform in (-2,2), so there is no need for the additional complexity of equation 9. Also, why would the temperature errors be uniformly distributed? why between -2 and 2 and which units represent those numbers (I guess C?). This is an example of a problem that goes through the whole manuscript.**

**Reply:** Thank you for your sincere and constructive suggestions. We used the perturbation method of meteorological forcing data in this study as described by Lei et al. (2014), which has been proven to be an effective method in data assimilation. In this study, the temperature unit is Kelvin, and we assumed the temperature error to be uniformly distributed. The variance scaling factor of the temperature was set to 2.0. We completely agree with the reviewer's opinion that equation 9 is unnecessarily complicated. If the reviewer and editor believe it to be unnecessary, we can certainly remove it. We have thoroughly revised the manuscript according to the reviewer's suggestions and appreciate the valuable feedback. Thank you.

Lei, F. N., Huang, C. L., Shen, H. F., et al. (2014), Improving the estimation of hydrological states in the SWAT model via the ensemble Kalman smoother: Synthetic experiments for the Heihe River Basin in northwest China, Advances in Water Resources, 67: 32-45.

**Response to comments by Reviewer #4:**

This paper optimizes the snow depth data simulated by the model by using the snow depth data observed at the station. The main questions are as follows:

**1. The title of the article is not accurate, and the purpose of assimilation cannot be obtained. The meaning of "snow" is too broad, so it needs to be specific;**

**Reply:** Thanks for your sincere and constructive suggestions. In this study, the Genetic Particle Filter was used as a data assimilation scheme to improve the simulation accuracy of snow depth by Noah-MP model, as the reviewer said, the meaning of "snow" is too broad in the title, and we have revised the title to "A genetic particle filter scheme for univariate data assimilation into Noah-MP model across snow climates", and we hope this title can highlight the theme of this manuscript.

**2. Is the Ws in NOAH-MP optimized by snow depth observation? Ws is the snow water equivalent. Is there any difference between the two?**

**Reply:** Thanks for your sincere and constructive suggestions. The snow water equivalent was calculated by snow depth and snow density in Noah-MP model, and we have noticed the snow water equivalent was optimized by snow depth observation. However, it is worth noting that most of the sites selected for this study do not have snow water equivalent observation data. Therefore, we were unable to include the assimilation results of snow water equivalent in the manuscript.

**3. The observation operator and model operator mentioned in Flow Chart 1 are not seen in the text, and need to be clarified;**
**Reply:** Thanks for your sincere and constructive suggestions. In this study, the Noah-MP model is the model operator mentioned in Flow Chart 1. In this study, the station observation data was assimilated into the Noah-MP model and there is no remote sensing observation data assimilated into the model. Therefore, the observation operator is an identity matrix in this study, and we have stated in section 2.

**4. The optimized state variables are not seen in the paper. If only the simulation results of NOAH-MP are corrected, which cannot be regarded as assimilation;**
**Reply:** Thanks for your sincere and constructive suggestions. The original objective of this study was to evaluate the performance of the genetic particle filter in different snow climates. To accomplish this, we assimilated observed snow depth at the point-scale into the Noah-MP model. As a result of the assimilation process, the snow simulation process was optimized, and we observed updates to all snow variables. However, we would like to note that due to the lack of snow water equivalent observation data at most of the sites, we were only able to present the assimilation results of snow depth in the manuscript.

**5. The calculation formula of fsnow, g needs to be given. Snow depth is also used in the calculation formula of fsnow, g. If snow depth is assimilated, has this been considered? If regional assimilation is carried out, how is fsnow, g calculated?**
**Reply:** Thanks for your sincere and constructive suggestions. The main objective of this study is to evaluate the performance of the Genetic Particle Filter across different snow climates, and to assess its feasibility as a point-scale snow data assimilation scheme. Through our study, we discovered that the fsnow variable can also be updated when observed snow depth is assimilated. Our next step is to expand this study to the regional level, where we will assimilate snow albedo products into the land surface model using the genetic particle filter to improve fsnow. In this study, our focus was on investigating the feasibility of GPF as a snow data assimilation scheme and evaluating its performance across different snow climates. We hope that the conclusions drawn from this manuscript will be a valuable reference for our upcoming regional snow data assimilation experiment.

**6. The introduction of assimilation process in this paper is not complete and detailed enough, and needs to be further improved.**
**Reply:** Thanks for your sincere and constructive suggestions. We have thoroughly revised the manuscript and provided a more detailed introduction to the assimilation process, as suggested by the reviewer.

**Response to comments by Reviewer #5:**
**The novelty of this research is not clear. If the genetic particle filter is a new method, then its advantages against other methods should be investigated directly. If the "across snow**

**climates" is a novelty, then I do not think the eight stations can represent snow climates considering the highly heterogeneous snow distribution. Finally, "the higher assimilating frequency, the higher simulation performance" is not surprising. Thus, I suggest the authors to clarify the novelty more clearly, so as to help the reader get the importance of this work. The method needs further introduction. Although the genetic particle filter data assimilation scheme is introduced in section 2.3, I am still confused that which variable you assimilate into the land surface model? If you only assimilate the snow depth, then how do you deal with other snow variables (e.g., snow water equivalent, snow density and snow age)? Also, how do you deal with the potential inconsistency between snow and ground temperature (for example, when the model shows no snow and the ground temperature is above zero, then how to assimilate the observed snow depth)? Actually, I am very concerned about the assimilation and evaluation. In figure 3, it seems you assimilate the observation every 5 days and then evaluate the model simulation at the same step? If this is the case, then will the direct insertion method show higher performance than the genetic particle filter?**

**Reply:** Thanks for your sincere and constructive suggestions. We have thoroughly revised the manuscript and provided a detailed introduction to the methodology. The primary aim of this study is to evaluate the feasibility of using the Genetic Particle Filter as a snow data assimilation scheme in point-scale and to examine its performance across different snow climates. To achieve this objective, we assimilated observed snow depth at point-scale into the Noah-MP model. Initially, the snow cover was divided into multiple layers based on the updated snow depth, and the snow water equivalent was updated using corresponding proportions calculated by the updated snow depth. Next, the snow depth was treated as an independent variable to calculate other snow variables, such as snow cover fraction, etc. During the snow period, we found that it could be divided into snow accumulation period, snow stable period, and snow ablation period. We only assimilated observed snow depth values that were above zero, that is, we did not assimilate observations when the model showed no snow. In this study, we assimilated snow depth observations every 5 days, while the temporal resolution of simulation results and snow depth observations used for evaluation was one day. Although we initially used the direct insertion method to improve the simulation results, we found that its performance was worse than the particle filter and could lead to a model shock during the assimilation period.

**The spatial difference. It seems the spatial difference among different stations is not strong and few information can be get (except the robust of the result, may be). Some insightful analysis on the spatial difference may help improve the manuscript. The writing needs careful proofreading.**

**Reply:** Thanks for your sincere and constructive suggestions. We have thorough revised the manuscript according to the reviewer's suggestion, and an English writing service has been purchased for this manuscript, thank you very much.

For example:
1. **L48: "succeeds in catching snow dynamics is" may be "succeed in catching snow dynamics is"**

**Reply:** Thanks for your sincere and constructive suggestions. We have revised in the manuscript.

2. **L51: "is aimed at investigating ... and obtain the ..." may be "is aimed at investigating ... and obtaining the ..."**

**Reply:** Thanks for your sincere and constructive suggestions. We have revised in the manuscript.

3. **L60: "However, this method possible result in ..." may be "However, this method possible results in ...".**
**Reply:** Thanks for your sincere and constructive suggestions. We have revised in the manuscript.

4. **L68: "this method does not require a model a model linearization." what do you mean?**
**Reply:** Thanks for your sincere and constructive suggestions. The Kalman Filter is a useful tool for linear systems, but it requires a model linearization for a nonlinear system. To overcome this limitation, the Monte Carlo approach was employed to approximate error estimates based on an ensemble of model simulations in the Ensemble Kalman Filter (EnKF). Consequently, the EnKF can be used in nonlinear systems without requiring model linearization.

**Response to comments by Reviewer #6:**
In this manuscript, You et al. set up a particle filtering framework using the Genetic Algorithm to avoid particle filter-typical degeneracy and sample impoverishment issues. They apply this framework to snow depth measuring sites in different climatological regions, expecting to learn about particle filter performance at each of the sites. They analyze the assimilated snow depth with respect to the suitability of their particle filtering algorithm for application in different snow climates, the influence of the used particle number on performance metrics, and the influence of assimilation window length on the performance. The manuscript is structurally well-organized. The topic is in general very interesting and the effort to push the field of data assimilation forward is very much appreciated. However, in my opinion there are some significant weaknesses in this manuscript:

**- a lacking motivation of the research question(s') relevance**
**Reply:** Thanks for your sincere and constructive suggestions. The original goal of this study was to investigate the performance of the Genetic Particle Filter as a snow data assimilation scheme across various snow climates. We attempted to resample the particles using a genetic algorithm. Following the reviewer's suggestions, we have thoroughly revised the manuscript.

**- a superficial description of the measurement sites and their properties, making a meaningful interpretation of the results with respect to the research question(s) difficult.**
**Reply:** Thanks for your sincere and constructive suggestions. We explored the sensitivity of snow simulations to parameterization schemes within the Noah-MP model using the same dataset of eight measurement sites as in our previous work (You et al., 2020). A detailed description, including the location map of the measurement sites, can be found in that paper. Therefore, in this manuscript, we have only briefly described the measurement sites. However, if the reviewer and editor feel that a more detailed description is necessary, we are happy to add it at any time. We have thoroughly analyzed the experimental results, as per the reviewer's suggestions. Thank you very much.

**- a lacking presentation of the Genetic Algorithm and stressing why this method is the most suitable for the analysis**
**Reply:** Thanks for your sincere and constructive suggestions. The degeneracy and sample impoverishment issues are common faults of particle filter, here, the genetic algorithm was used in resampling process and was expected to effectively mitigate these issues. We have made a detailed

description of the Genetic Particle Filter in the manuscript according to the reviewer's suggestion, thank you very much.

**- a superficial interpretation of the results, in particular with respect to the overarching hypothesis (different filter performance in different "snow climates")**

**Reply:** Thanks for your sincere and constructive suggestions. We have thorough revised the manuscript and presented a detailed discussion in the manuscript according to the reviewer's suggestion.

**- issues with the used literature in the References section and in general a rather scarce literature selection. Difficulties in the use of English, which makes some sections of the manuscript hard to understand**

**Reply:** Thanks for your sincere and constructive suggestions. We have completed the references and thorough revised the manuscript according to the reviewer's suggestion. Additionally, an English writing service has been purchased for this manuscript. Thank you very much.

**- an intransparent (or simply not listed?) choice of model parameter values and meteorological values to perturb; unclear or not explained error distribution choices.**

**Reply:** Thanks for your sincere and constructive suggestions. In this study, the model parameters were obtained from the look-up table within the Noah-MP model, based on the soil and vegetation type of the sites. We did not perturb the model parameters but rather perturbed the meteorological forcing to produce ensembles. The perturbation method and the error distribution choices were presented in the manuscript.

**- a results and discussion section that partly loses contact with the research questions**

**Reply:** Thanks for your sincere and constructive suggestions. We acknowledge that the discussion section in the previous version lacked connection with the research questions, and we have thoroughly revised the manuscript based on the reviewer's suggestions.

If this manuscript is accepted for a major revision process, it should be largely rewritten and then undergo line-by-line comments in a second review iteration. The focus should first be on the following aspects:

**- reworking the manuscript research questions (is it about the filter performance in different climates as the title suggests or about the three questions formulated at the end of the introduction, or both?)**

**Reply:** Thanks for your sincere and constructive suggestions. We have thorough reworked the manuscript according to the reviewer's suggestion.

**- a more comprehensive literature review on the technical literature regarding the research questions**

**Reply:** Thanks for your sincere and constructive suggestions. We have thorough revised the manuscript according to the reviewer's suggestion.

**- a more detailed description of the used particle filter method and why this filter is chosen to be the most suitable to answer the research question**

**Reply:** Thanks for your sincere and constructive suggestions. We have provided a comprehensive explanation of the genetic particle filter process in our manuscript and compared its assimilation results to those obtained using particle filters with generic resampling methods in section 3.5.

**- a more critical questioning of the results, in particular with respect to the 100-particle threshold (e.g. why in Fig. 7 a minimum exists at 100 particles).**

**Reply:** Thanks for your sincere and constructive suggestions. In this study, we performed a sensitivity analysis of the Genetic Particle Filter (GPF) to particle number at eight sites. Our findings show that the root mean square error (RMSE) of the assimilation results decreased as the particle number increased. We observed that the RMSE reached a minimum value when the particle number was equal to 100. However, increasing the number of particles beyond 100 led to an increase in RMSE. In other words, 100 particles were sufficient to represent the model state. Excessive particles led to a poor assimilation result and increased computational burden.

**Response to comments by Editor:**

Dear authors,

This manuscript uses field data to improve snow modelling through the genetic particle filter algorithm. The topic is highly relevant and fits nicely within this SI. However, the manuscript requires significant improvements in order to be considered for possible publication in HESS. The Introduction section should be shorted, more concise and better highlight the research question and the need for this study, and use more updated references. The study sites must be better described, namely regarding the snow differences between the sites, so that we can understand the real application of your proposed method. The methodology requires relevant improvements to describe the genetic particle filter algorithm, and all the mathematical assumptions performed. The Results must be clearly presented and discussed. Discussion should clearly present the advantages of the proposed method comparing with others, and the limitations linked with the assumption performed. It is also important to compare the results with those from previous studies, and bring more references into this section. The Conclusions section must clearly identify the novelty and main messages of this study, and clearly identify why the proposed method is better than the available ones. Language editing is also required. More detailed comments have been provided by reviewers and must be considered in improving the manuscript. There was one late reviewer whose comments are provided bellow and should be also considered.

**Reply:** Thanks for your sincere and constructive suggestions. As you said, some significant weaknesses in this manuscript and we have thorough revised the manuscript according to the Editor and Reviewer's suggestion, thank you very much. For the language editing, we have purchased an English writing service for this manuscript. And we look forward to receiving your positive feedback this time. Thank you very much.

---

## Author Response (AR2)

**Response to Comments on the Manuscript:**

**"Investigating the performance of Genetic Particle Filter in snow data assimilation across snow climates"**

**(HESS-2022-350)**

**May 17, 2023**
* * *
The authors gratefully acknowledge the editors and the anonymous reviewers for their constructive comments. We have made a comprehensive revision of our previous manuscript. The main modifications have been highlighted in yellow color in order to highlight the issues raised by the editors and anonymous reviewers, and are summarized as follows:

For more details, please refer to the item-by-item response. Thank you for your time.

**Response to comments by Reviewer #2:**
We would like to take this opportunity to gratefully thank the reviewer for his/her constructive comments and recommendations for improving the paper. An item-by-item, point-by-point response to the interesting comments raised by the reviewer follows.

Main points:

**1) The manuscript still needs a thorough copy editing. I have included suggestions on grammatical corrections below, but I have stopped at about page 6, since the text contained errors almost every other line. It can be cumbersome for foreign authors, like myself, but nowadays, several software packages assist the authors. In any case, a manuscript should appear written in proper English. Furthermore, it should be the task of a reviewer to correct the grammar of a manuscript, even less so when this is the second version.**

**Response:** Thank you for your sincere and constructive suggestions. Despite our best efforts in revising this manuscript, it may still contain some errors. We would like to express our gratitude to the reviewer for correcting the grammar and syntax issues present in the manuscript. Additionally, we have thoroughly proofread the entire document and even utilized the services of a senior English writing expert to ensure that all language and grammar issues have been addressed.

**2) As with the first version, the data assimilation's real purpose in the snow depth context**

remains unclear. Usually, a model assimilates a set of limited observations but provides a much more complete output regarding variable and spatial coverage. For instance, a weather prediction model assimilates available station observations of some variables but produces a complete field of future predictions. Here, the hydrological model assimilates snow depth observations but estimates snow depth at the same location, and it does not even produce predictions of snow depth. So the utility of the whole set-up is unclear. Shouldn't the model be validated by comparing prediction versus observations? The conclusion of the study that the results of assimilation are closer to observations is to ma a tautology. Of course., the model results are corrected towards the observations. It is no surprise that those corrected values are closer to observations. A natural progress would be if the model \*predictions\* for a specific time step t are closer to observations when the observations prior to t are assimilated.

**Response:** Thank you for your sincere and constructive suggestions. We strongly endorse the reviewer's opinion that a set of limited observations assimilated by a model should lead to a comprehensive output. The original goal of this study was to investigate the performance of the Genetic Particle Filter as a snow data assimilation scheme across various snow climates. The model was driven by meteorological forcing data and halted when the observation occurred, then the observation data was assimilated into the model. Here, the observed snow depth at the point-scale was assimilated into Noah-MP model and the assimilation step was set to five days. Our findings demonstrate a noticeable improvement in the model results after each assimilation step. While we agree with the reviewer's suggestion of validating the model predictions against actual observations, it must be noted that in this study, historical data was used to perform experiments with assimilations being carried out every five days, resulting in only five-day-long "model predictions". Therefore, the effectiveness of the assimilation approach was evaluated by comparing the assimilation results, non-assimilated model simulations, and observations.

Particular points:

**3) line 78 'and AMSR-E SWE into a hydrologic model to improve modeled SWE..' hydrological model.**

**Response:** Thank you for your sincere and constructive suggestions. We have revised in the manuscript and highlighted in yellow color.

**4) line '81 assimilating ground-based snowfall and snowmelt rates, simultaneous assimilation of D-InSAR. The acronym has not been defined yet.**

**Response:** Thank you for your sincere and constructive suggestions. We have revised in the manuscript and highlighted in yellow color.

**5) line 88 "dynamic system" usually has strong nonlinearity. change to "dynamical system".**

**Response:** Thank you for your sincere and constructive suggestions. We have revised in the manuscript and highlighted in yellow color.

**6) line 95 "The greatest strength of PF technique is free from the constraints of model linearity and error following Gaussian distribution, this makes the PF technique succeed applied in nonlinear and non-Gaussian dynamic systems." change to "the greatest strength of PF technique is to be free from the constraints of model linearity and error following a Gaussian distribution. This allows the successful application of the PF technique to nonlinear dynamical systems with non-Gaussian errors".**

**Response:** Thank you for your sincere and constructive suggestions. We have revised in the manuscript and highlighted in yellow color.

**7) line 97 "dynamic systems. Additionally, PF technique give weights" change to "dynamical systems. Additionally, the PH technique gives weights."**

**Response:** Thank you for your sincere and constructive suggestions. We have revised in the manuscript and highlighted in yellow color.

**8) line 108 "snowpack runoff simulations (Magnusson et al., 2017). Above studies demonstrated that either assimilated the snow-related in-situ measurements or remotely sensed observation data through PF technique can successfully update the predictions of snowpack dynamics," change to "The studies indicated above demonstrated that the assimilated snow-related in-situ measurement or the remotely sensed observation data through PF technique can successfully update the predictions of snowpack dynamics."**

**Response:** Thank you for your sincere and constructive suggestions. We have revised in the manuscript and highlighted in yellow color.

**9) line 112 "Nevertheless, particle degeneracy is still one potential limitation for PF technique, it occurs when most of particles have negligible weight and only few particles have significant weights, which makes the state probability distribution cannot be represented by the particles" to "Nevertheless, particle degeneracy is still one potential limitation of PF technique. It occurs when most particles have negligible weight, and only a few particles carry significant weights, which hinders a realistic sampling of the underlying probability distribution of the state."**

**Response:** Thank you for your sincere and constructive suggestions. We have revised in the manuscript and highlighted in yellow color.

**10) line 116 "an efficient approach which can effectively mitigate the problem of particle degeneracy, however, may lead to the resulting sample will contain many repeated points a"**

change to **"An efficient approach that can effectively mitigate the problem of particle degeneracy. However, it may lead to the resulting sample containing many repeated points a."**

**Response:** Thank you for your sincere and constructive suggestions. We have revised in the manuscript and highlighted in yellow color.

**11) line 147 is distributed at different latitudes in the "northern hemisphere" change to "Northern Hemisphere."**

**Response:** Thank you for your sincere and constructive suggestions. We have revised in the manuscript and highlighted in yellow color.

**12) line 148 "located beside the Kitinen River in Finland and has a 2 m depths frost" change to "located beside the Kitinen River in Finland. The upper 2 meters are frozen."**

**Response:** Thank you for your sincere and constructive suggestions. We have revised in the manuscript and highlighted in yellow color.

**13) line 164 It is noteworthy that "the spatial variance on the performance of the model" is negligible change to "the spatial variance of the performance of the model".**

**Response:** Thank you for your sincere and constructive suggestions. We have revised in the manuscript and highlighted in yellow color.

**14) line 168 detailed information of snow climates, and dataset process introduction of the eight sites can be also referenced in You et al. (2020a). What is 'data process'. ....can also be found in You et al. (2020a).**

**Response:** Thank you for your sincere and constructive suggestions. Regarding the dataset process, it involves the processing of original meteorological measurements from eight sites for the experiment. For instance, some subhourly measurements were converted to hourly at certain sites. You can find more information on the data processing method in You et al. (2020).

**15) line 170 The snow partial within Noah-MP model. This is not proper English. It is unclear what snow partial is.**

**Response:** Thanks for your sincere and constructive suggestions. Based on your feedback, we have revised 'snow partial' to 'snow partial module' in the manuscript and highlighted in yellow color. This term refers to the snow module within the Noah-MP model.

**16) line 213 function p ( zt xt i ) , which measures the likelihood of a given model state concerning the observation z t. The notation could be clearer. Usually, I would interpret p (z_t) | x_ti) as the probability of z_t conditional on x_ti.**

**Response:** Thanks for your sincere and constructive suggestions. Typically, the $p\left(x_t^i \big| z_t\right)$ denotes the probability of $x_t^i$ conditional on $z_t$, which is referred to as the posterior probability of $x_t^i$. Likewise, the $p\left(z_t \big| x_t^i\right)$ denotes the probability of $z_t$ conditional on $x_t^i$, known as the likelihood probability. However, if the reviewer feels that a clearer notation is necessary, we are happy to change it at any time.

**17) line 215 "In general, a Gaussian distribution was assumed to perturb the observations and the likelihood function was defined to represent the errors." change to "The observation errors are generally assumed to follow a Gaussian distribution, and the chosen likelihood function represents this assumption."**

**Response:** Thanks for your sincere and constructive suggestions. We have revised in the manuscript and highlighted in yellow color.

**18) line 225 if the effective sample size. what is the effective sample size? Is it just the number of samples? The text does not mention autocorrelation at all, so the word 'effective' is unclear.**

**Response:** Thanks for your sincere and constructive suggestions. As mentioned in the manuscript, particle filter schemes suffer from the degeneracy phenomenon. In fact, after several iterations, all but one particle will have negligible weight. To measure the degree of degeneracy, the effective ensemble size $N_{eff}$ is a suitable metric. In our case, the estimation of $N_{eff}$ can be calculated by $N_{eff} = 1 \Big/ \sum_{i=1}^{N} \left(w_t^i\right)^2$, and a small value indicates severe degeneracy (Mechri et al., 2014; Piazzi et al., 2018).

**19) line 242 The role of the survival rate is unclear. It seems that the survival rate is just a measure of the distance between the particle and the observations. This distance is already considered when assimilating the observations with the weighted average over particles. So is this a double counting?**

**Response:** Thanks for your sincere and constructive suggestions. The Genetic Algorithm could be defined as a stochastic searching algorithm (a function optimizer) ensuing from Darwin's evolution theory, simulating the well-known *survival of the fittest* evolution. In this study, the measurement of fitness was determined by the survival rate, which was calculated based on the distance between the particles and observations to select high-quality particles. Additionally, the distance was also considered when assimilating the observations, in order to update the weight of particles. Consequently, this does not constitute double counting, as noted by the reviewer.

**20) line 268 All particles are disturbed with a gaussian error. Isnt is just the same as the mutation, only with a different type of error distribution? what is the role of the mutation?**

**Response:** Thank you for your sincere and constructive suggestions. Our approach involves using a complete GA (genetic algorithm) that re-supplies or re-defines particles through the selection, crossover, and mutation operators. The mutation operator plays a crucial role in increasing particle diversity and avoiding particle impoverishment. In this study, we implemented the mutation process using equation (11) and assumed a random number from a uniform distribution. Since the particles represent the model variable 'snow depth' and must be greater than or equal to zero, introducing a random number from a Gaussian distribution may result in a negative particle value, which would cause the model to stop.

**21) line 339 'Since the meteorological perturbations are unbiased, the nonlinearity of physical processes within the model is supposed to be the main reason for the uncertainty'. I may not understand this sentence. The magnitude of uncertainties is not related to the linear or nonlinear character of the model. A linear model would just rescale the spread in the forcing and nonlinear model would expand or shrink disproportionly the forcing uncertainties. So the nonlinear character itself cannot be the reason per se of.**

**Response:** Thank you for your sincere and constructive suggestions. We greatly appreciate the reviewer's opinion that the magnitude of uncertainties is not solely dependent on the linear or nonlinear character of the model. In our view, the model structure is one of the main reasons for uncertainty, and it is influenced by the degree of complexity of physical processes with nonlinear characteristics. And we have revised in the manuscript and highlighted in yellow color.

**22) Section 3.1 Open-loop ensemble simulations. The expression open loop-ensemble is used only once in the title of this section. What is its meaning? It is nowhere defined nor used again. This section is also tough to read. It contains just one very long paragraph without clear structure.**

**Response:** Thank you for your sincere and constructive suggestions. The open-loop ensemble simulations mean the ensemble simulations forced by perturbed meteorological data and without data assimilation. With the aim of properly analyzing the skill of the data assimilation scheme, the assimilation results are evaluated through comparison with the control open-loop. We have revised the manuscript and presented a clear structure.

Mechri, R., Ottle, C., Pannekoucke, O., and Kallel, A.: Genetic particle filter application to land surface temperature downscaling, Journal of Geophysical Research-Atmospheres, 119, 2131-2146, 2014.

Piazzi, G., Thirel, G., Campo, L., and Gabellani, S.: A particle filter scheme for multivariate data assimilation into a point-scale snowpack model in an Alpine environment, Cryosphere, 12, 2287-

2306, 2018.

You, Y. H., Huang, C. L., Yang, Z. L., Zhang, Y., Bai, Y. L., and Gu, J.: Assessing Noah-MP parameterization sensitivity and uncertainty interval across snow climates, Journal of Geophysical Research-Atmospheres, 125, e2019JD030417, 2020.